# Vγ1 and Vγ4 gamma-delta T cells play opposing roles in the immunopathology of traumatic brain injury in males

Hadi Abou-El-Hassan [1,3], Rafael M. Rezende[1,3], Saef Izzy [1], Galina Gabriely[1], Taha Yahya[1], Bruna K. Tatematsu[1], Karl J. Habashy [1], Juliana R. Lopes[1], Gislane L. V. de Oliveira [1], Amir-Hadi Maghzi[1], Zhuoran Yin[1], Laura M. Cox [1], Rajesh Krishnan[1], Oleg Butovsky [1,2] & Howard L. Weiner [1] ✉

Traumatic brain injury (TBI) is a leading cause of morbidity and mortality. The innate and adaptive immune responses play an important role in the pathogenesis of TBI. Gamma-delta (γδ) T cells have been shown to affect brain immunopathology in multiple different conditions, however, their role in acute and chronic TBI is largely unknown. Here, we show that γδ T cells affect the pathophysiology of TBI as early as one day and up to one year following injury in a mouse model. TCRδ$^{-/-}$ mice are characterized by reduced inflammation in acute TBI and improved neurocognitive functions in chronic TBI. We find that the Vγ1 and Vγ4 γδ T cell subsets play opposing roles in TBI. Vγ4 γδ T cells infiltrate the brain and secrete IFN-γ and IL-17 that activate microglia and induce neuroinflammation. Vγ1 γδ T cells, however, secrete TGF-β that maintains microglial homeostasis and dampens TBI upon infiltrating the brain. These findings provide new insights on the role of different γδ T cell subsets after brain injury and lay down the principles for the development of targeted γδ T-cell-based therapy for TBI.

Traumatic brain injury (TBI), known as the silent epidemic[1], is a complex neurologic process that includes acute molecular changes and long-term neurocognitive sequelae[2,3]. TBI-related emergency department visits, hospitalization, and deaths increased by 53% from 2006 to 2014 in the United States posing a pressing public health concern[4]. TBI is a debilitating neurological condition due to its significant association with post-traumatic functional impairment and risk for Alzheimer's disease[5,6], amyotrophic lateral sclerosis[7], chronic traumatic encephalopathy[8], epilepsy[9], depression[10], post-traumatic stress disorder[11], and chronic pain[12]. These diseases are the result of an array of cellular and subcellular events that include changes in glia following brain injury that leads to neuroinflammation[13–15]. There is no current treatment that targets the neuroinflammatory process in TBI, in part because the cellular and molecular mechanisms leading to

neurological deficits after TBI are unknown[16,17]. Glial cells including microglia, are resident cells of the central nervous system (CNS) that play a pivotal role in maintaining neuroglial homeostasis[18,19] and neurovascular integrity[20]. Early microglial responses following TBI may have restorative effects whereas persistent activation is linked to progressive neurodegeneration[21–23]. Microglial activation occurs rapidly after both animal[24] and human TBI[25]. Consistent with this, TBI in aging cohorts, in which microglia are overactivated, results in more severe neuronal damage and functional deficits in both experimental and human studies[26].

Infiltrating immune cells including monocytes and lymphocytes are known to contribute to TBI pathogenesis[27]. In this context, gamma-delta (γδ) T cells have emerged as potential players in neuroinflammation[28,29]. γδ T cells are a subset of lymphocytes bearing a T cell receptor

[1]Ann Romney Center for Neurologic Diseases, Brigham & Women's Hospital, Harvard Medical School, Boston, MA, USA. [2]Evergrande Center for Immunologic Diseases, Brigham and Women's Hospital, Harvard Medical School, Boston, MA, USA. [3]Thsese authors contributed equally: Hadi Abou-El-Hassan, Rafael M. Rezende. ✉e-mail: hweiner@rics.bwh.harvard.edu

composed of gamma (γ) and delta (δ) chains as opposed to alpha (α) and beta (β) chains found on conventional CD4 and CD8 T cells[30]. γδ T cells play a critical role in mediating both innate[31] and adaptive[32] immune responses. They are considered part of the first line of defense against pathogens as they can rapidly respond to T cell receptor (TCR) signals in an MHC-independent manner[30] and to pattern recognition receptors signals such as TLR[33]. γδ T cells are more abundant in non-CNS compartments such as lymphoid organs[34], skin[35] and intestines[36] as well as in peri-CNS compartments such as blood[37] and meninges[38,39]. The repertoire of γδ T cells is determined by the types of Vγ and Vδ chain combinations expressed at their TCR[30]. Vγ1 and Vγ4 γδ T cells (nomenclature based on Heilig and Tonegawa[40]) are the only γδ T cell subsets that are continuously generated in the thymus after birth and can secrete a variety of cytokines including IFN-γ, IL-17 and TGF-β[41,42]. These γδ T cell subsets circulate among lymphoid tissues to survey against pathogens and malignant cells[43,44]. Characterization of γδ T cells in the CNS is poorly understood and the contributions of γδ T cells to the pathogenesis of TBI have not been explored.

In this study, we find that Vγ1 and Vγ4 subsets of γδ T cells play opposing roles in the pathogenesis of TBI. While Vγ1 γδ T cells are protective due to the secretion of TGF-β, Vγ4 γδ T cells potentiate inflammation following TBI by secreting IFN-γ and IL-17. Importantly, these effects are dependent on the crosstalk between infiltrating γδ T cells and microglia in which TGF-β favors a homeostatic microglial gene signature whereas IFN-γ and IL-17 favor an inflammatory microglial phenotype. Our studies demonstrate that γδ T cell subsets play a key role in TBI and shed new light into therapeutic approaches that target γδ T cells.

## Results

### γδ T cells contribute to CNS damage and functional deficits following TBI

To investigate whether γδ T cells play a role in TBI, we employed the controlled cortical impact (CCI) model of TBI using a stereotaxic electromagnetic impactor. We induced moderate TBI over the right parietal cortex of TCRδ⁻/⁻ and age-matched C57BL6/J wild-type (WT) mice. Hematoxylin and eosin (H&E) staining of the brain 7 days after TBI showed significant reduction in lesion volume in the TCRδ⁻/⁻ mice compared to WT (Fig. 1a, b). We also found significant reduction in brain edema of the ipsilateral hemisphere in the TCRδ⁻/⁻ group at 3 days post-CCI (Fig. 1b). To investigate the effect of γδ T cells on motor function, we performed the rotarod test[45] and found improvement in motor function 2 weeks after TBI in TCRδ⁻/⁻ mice (Fig. 1c). We then investigated post-traumatic anxiety using the open field test[46,47] and found that at 1 month after TBI, TCRδ⁻/⁻ mice exhibited increased exploratory behavior and reduced anxiety-like behavior (Fig. 1d, e), increased traveling velocity (Fig. 1f), increased total distance traveled (Fig. 1g) and increased immobility in the central zone of the open field arena (Fig. 1h). One year after injury, TCRδ⁻/⁻ mice had a complete restoration of memory function (Fig. 1i, j), increased time spent in the target quadrant during the probe trial (Fig. 1k), increased target quadrant transitions (Fig. 1l) and increased swimming velocity (Fig. 1m) as observed in the Morris water maze (MWM) test[48,49]. However, the marble burying test, which is used to evaluate, compulsive behavior in TBI[50], showed that TCRδ⁻/⁻ mice had increased compulsive behavior one year post-TBI (Fig. 1n), suggesting that γδ T cells play a beneficial role in TBI-induced compulsive behavior. Given the recent evidence on divergent sex responses at the molecular[51] as well as behavioral[52] levels after TBI[53], we investigated whether female TCRδ⁻/⁻ animals confer comparable behavioral neuroprotection to their male counterparts. We found no significant differences at the behavioral level between WT-TBI and TCRδ⁻/⁻ TBI female mice at 1 month after injury (Supplementary Fig. 1a). We also found that, except for anxiety-like behavior, TBI itself did not significantly affect cerebellar function or memory between WT-Sham and WT-TBI female mice. This may be explained by the dampened acute neuroinflammation in

females that has been proposed to be driven by the neuroprotective effects of estrogen and progesterone[54].

We have previously found that the gut microbiota differ in TCRδ⁻/⁻ mice vs. WT controls[55]. Consistent with our prior studies, we detected changes in an independent cohort of TCRδ⁻/⁻ and WT mice (Supplementary Fig. 1b–e). We performed 16 S rRNA sequencing and confirmed that the microbiota differs in WT vs. TCRδ⁻/⁻ mice in β-diversity (PERMANOVA $p < 0.001$ based on weighted and unweighted UniFrac distances), and in relative abundance of specific bacteria. Interestingly, *Akkermansia muciniphila*, one of the well-studied bacteria in the gut-brain axis was found to be enriched in TCRδ⁻/⁻ mice. *Akkermansia muciniphila*, hypothesized to play a protective role[56], was also enriched in an animal model of Alzheimer's Disease[56] as well as in patients with intracranial hemorrhage[57].

Take together, these findings demonstrate the critical role of γδ T cells on neurocognitive functions after TBI.

### γδ T cells drive CNS inflammation in TBI

To investigate the role of γδ T cells in acute CNS inflammation in TBI, we isolated immune cells from brains of TCRδ⁻/⁻ and WT mice (Fig. 2a). At 2 days post-TBI, TCRδ⁻/⁻ mice showed reduction of CD4+ (Fig. 2b) and CD8+ (Fig. 2c) T cells, CD11b+Ly6Chi classical monocytes (Fig. 2d), CD11b+CD45hiCX3CR1+CCR2+ monocytes[58] (Fig. 2e), and CD11b+Ly6G+ neutrophils (Supplementary Fig. 2a) as well as reduced MHCII expression on CD45int microglia/macrophages (Fig. 2f). Because microglial cells are known to play an important role in neuroinflammation following TBI[24,25,59], we asked whether microglia were affected by the absence of γδ T cells. We found reduced colocalization of iNOS on Iba1+ microglia in the pericontusional cortex of TCRδ⁻/⁻ mice (Fig. 2g; Supplementary Fig. 2b) as well as increased ramifications of Iba1+ microglia using Scholl analysis (Fig. 2h), suggesting that γδ T cells play a role in microglial activation.

Increased APP immunoreactivity and Tau phosphorylation (p-Tau) have proven to be critical characteristics of neuroinflammation following TBI[60]. At 7 days after TBI, WT mice demonstrated increased APP and p-Tau immunoreactivity (Fig. 2i) whereas TCRδ⁻/⁻ mice had less APP and p-Tau immunoreactivity (Fig. 2i). Consistently, RT-qPCR analysis of total brain RNA lysate from TCRδ⁻/⁻ mice 2 days after injury showed reduction of key inflammatory cytokines including *Il12a*, *Il18*, *Il6*, *Tnf*, *Il1b*, *Ccl2*, *Il17a*, *Cxcl10* and *Ccl5* whereas the regulatory T cell (Treg) master transcription factor *Foxp3* was increased in TCRδ⁻/⁻ mice (Fig. 2j; Supplementary Fig. 2c). Thus, γδ T cells play a major role in CNS inflammation following TBI and modulate both innate and adaptive immunity.

### γδ T cells modulate microglial phenotype in acute and chronic TBI

Single-cell suspensions of microglia were obtained following TBI using the microglia-specific 4D4+ antibody developed by our laboratory[61,62]. We analyzed microglial phenotypes in three comparisons: TBI vs. sham, TCRδ⁻/⁻ vs. WT, and aged vs. young (Supplementary Fig. 3a–e; Supplementary Data 1). PCA analysis of the microglial transcriptome in the acute phase (2 days post-injury; young) and in the chronic phase (1 year post-injury; aged) of TBI revealed distinct signatures (Fig. 3b). We detected 20,212 expressed genes and found 2437 differentially expressed genes (DEG) ($P < 0.05$) in microglia isolated from young acute WT-TBI mice vs. young WT-Sham (Fig. 3c–e). Gene set enrichment analysis (GSEA) revealed that these transcripts were associated with inflammatory biological pathways including IFN-γ response, apoptosis, and TNF-α signaling (Supplementary Fig. 3f), which is consistent with a previous report[63]. Compared to young acute WT-TBI, we detected 1062 DEGs in microglia from young acute TCRδ⁻/⁻ TBI mice (Fig. 3f). In acute TBI, microglia from TCRδ⁻/⁻ mice had a less inflammatory and a more homeostatic gene signature. Homeostatic genes including *Tmem119*, *Lrrc3* and *Siglech* were increased whereas

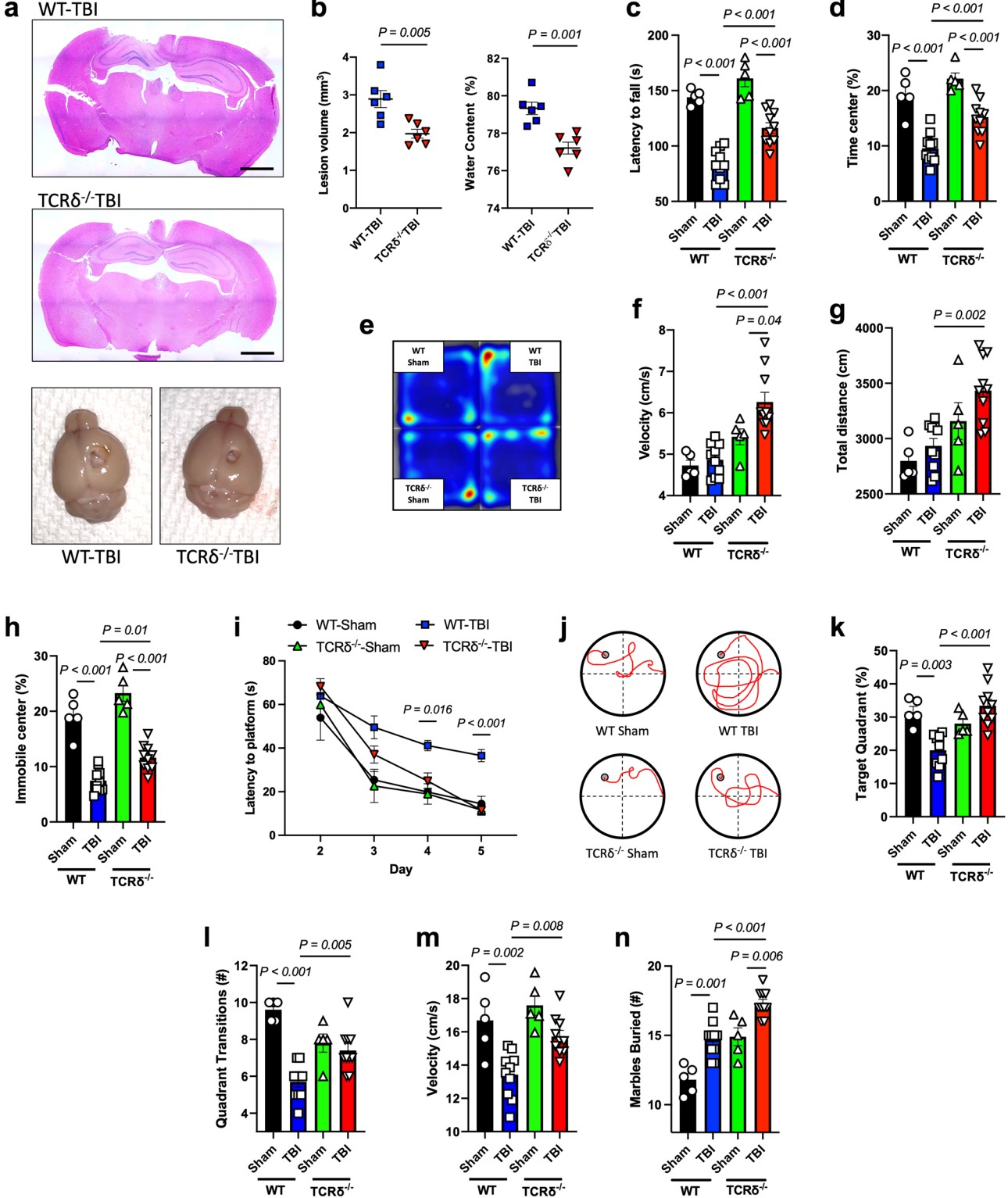

**Fig. 1 | γδ T cells contribute to functional deficits in TBI. a** H&E staining and representative pictures of the TBI lesion. **b** Lesion volume (left) of WT and TCRδ⁻/⁻ mice 7 days after TBI (WT-TBI $n = 6$, TCRδ⁻/⁻ TBI $n = 6$) and brain edema (right) at 3 days after TBI (WT-TBI $n = 6$, TCRδ⁻/⁻ TBI $n = 6$). **c** Rotarod test performed two weeks after TBI showing the latency time to fall (WT-Sham $n = 5$, WT-TBI $n = 10$, TCRδ⁻/⁻ Sham $n = 5$, TCRδ⁻/⁻ TBI $n = 10$). **d–h** Open field test performed one month after TBI showing the percentage of time spent in the center of the field **d** with the corresponding heatmap **e**, total velocity **f**, total distance traveled **g**, and the percentage of time of immobility in the center of the field **h** (WT-Sham $n = 5$, WT-TBI $n = 10$, TCRδ⁻/⁻ Sham $n = 5$, TCRδ⁻/⁻ TBI $n = 10$). **i–m** Morris water maze test performed one year after TBI showing the time latency to platform **i** and the corresponding trajectory **j**, percent of time spent in the target quadrant during the probe trial **k**, the number of total target quadrant transitions during the probe trial **l** and the total velocity **m** (WT-Sham $n = 5$, WT-TBI $n = 10$, TCRδ⁻/⁻ Sham $n = 5$, TCRδ⁻/⁻ TBI $n = 10$). **n** Marble burying test performed one year after TBI showing the total number of marbles buried (WT-Sham $n = 5$, WT-TBI $n = 10$, TCRδ⁻/⁻ Sham $n = 5$, TCRδ⁻/⁻ TBI $n = 10$). Biologically independent samples were used. Data shown as mean ± sem. $P$ values calculated by two-tailed unpaired Student's $t$-test **b**, one-way ANOVA with Bonferroni's multiple comparison test **c**, **d**, **f–h**, **k–n** and two-way repeated measures ANOVA with Bonferroni's multiple comparison test **i**. WT wild-type, TBI traumatic brain injury, TCR T cell receptor. Scale bars: **a** 2 mm. Source data are provided as a Source Data file.

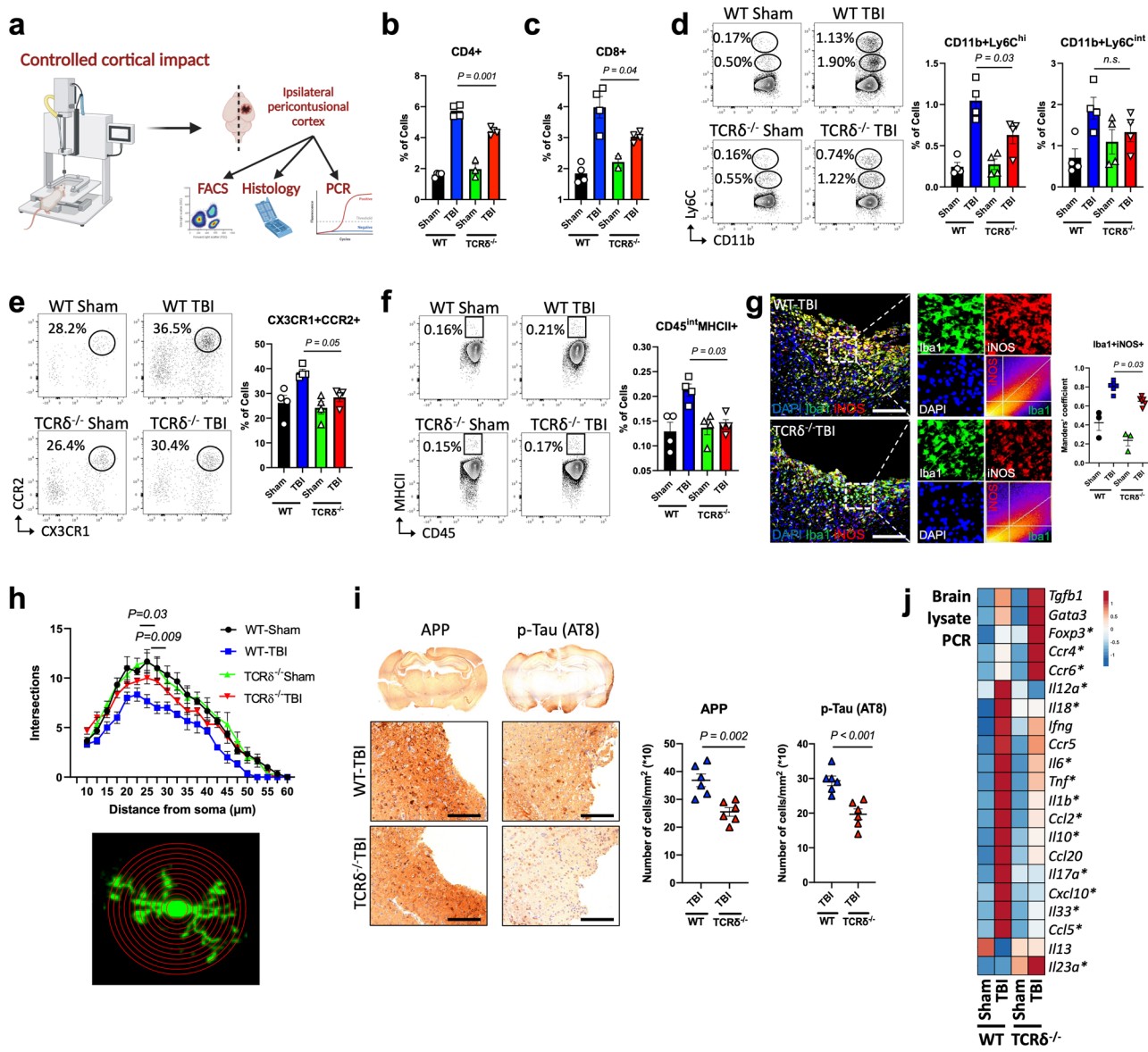

**Fig. 2 | γδ T cells drive acute neuroinflammation after TBI. a** Schematic of the experimental layout. Mice were subject to TBI using the CCI model and the peri-contusional brain tissue was harvested. Created with BioRender.com. **b–f** Flow cytometric quantification of brain CD3 + CD4 + T cells **b**, CD3 + CD8 + T cells **c**, CD11b+Ly6C[hi] and CD11b+Ly6C[int] **d**, CD11b+CD45[hi]CX3CR1 + CCR2+ **e**, and CD45[int]MHCII+ myeloid cells **f** (WT-Sham $n = 4$, WT-TBI $n = 4$, TCRδ[-/-] Sham $n = 4$, TCRδ[-/-] TBI $n = 4$). **g** Immunofluorescence staining (left) and quantification (right) of pericontusional brain samples for Iba1 (green), iNOS (red) and DAPI (blue). Dotted boxes showing a focused area of interest along with colocalization analysis of the overlap of Iba1 and iNOS (WT-Sham $n = 3$, WT-TBI $n = 6$, TCRδ[-/-] Sham $n = 3$, TCRδ[-/-] TBI $n = 6$). **h** Sholl's analysis of microglial ramifications (WT-Sham $n = 3$, WT-

TBI $n = 6$, TCRδ[-/-] Sham $n = 3$, TCRδ[-/-] TBI $n = 6$) with $P$ values corresponding to TCRδ[-/-] TBI versus WT-TBI at the corresponding distance from soma. **i** Bright field staining of pericontusional brain samples for APP and p-tau (AT8) (WT-TBI $n = 6$, TCRδ[-/-] TBI $n = 6$) performed at seven days after injury. **j** Quantitative RT-qPCR of cytokines and factors measured from brain lysate at two days after TBI where (*) indicates statistical significance $P < 0.05$ for TCRδ[-/-] TBI versus WT-TBI (see Supplementary Fig. 2c for quantification). Biologically independent samples were used. Data shown as mean ± sem. $P$ values calculated by one-way ANOVA with Bonferroni's multiple comparison test **b–h** and two-tailed unpaired Student's $t$-test **i**. APP: amyloid precursor protein; p-tau: phosphorylated-tau. Scale bars: **g**, **i** 70 μm. Source data are provided as a Source Data file.

inflammatory microglia genes including *Ccl2, Casp1, Tlr4, Il1a, Cd86, Clec7a, Hmgb1* and *Prdx1* were reduced[61] (Fig. 3c–f). GSEA revealed suppression of biological pathways involved in TNF-α signaling, apoptosis, protein secretion, IL-6/JAK/STAT3 signaling and MTORC1 signaling in microglia isolated from young acute TCRδ[-/-] TBI mice (Fig. 3g, h). An interaction cnetplot between the acute pathways altered in TCRδ[-/-] TBI microglia is shown in Fig. 3i. Acute microglia modulation was predicted to be driven by both IFN-γ ($z = -1.765$) and IL-17A ($z = -1.965$) as upstream regulators (Fig. 3j; Supplementary Fig. 3g, h). IFN-γ and IL-17A were selected among all predicted regulators since γδ T cells are an important source of these cytokines.

In the chronic phase of TBI, we detected 1376 DEGs that were differentially regulated ($P < 0.05$) in microglia from aged WT-TBI mice vs. aged WT-Sham (Fig. 3k, m). These DEGs were associated with enrichment in biological pathways including TGF-β signaling and reduced apoptosis (Supplementary Fig. 3i). Compared to aged chronic WT-TBI mice, we detected 59 differentially expressed genes in microglia isolated from aged chronic TCRδ[-/-] TBI mice (Fig. 3k, l, n). In the absence of γδ T cells at 1 year after injury, the level of expression of microglia genes including *Hspa1a* (involved in heat-shock response[64]), *Tram1* (promotes M1-polarization[65]), *Rnf187* (a c-Jun co-activator[66]) and *Stip1* (a stress-induced phosphoprotein that accelerates amyloid

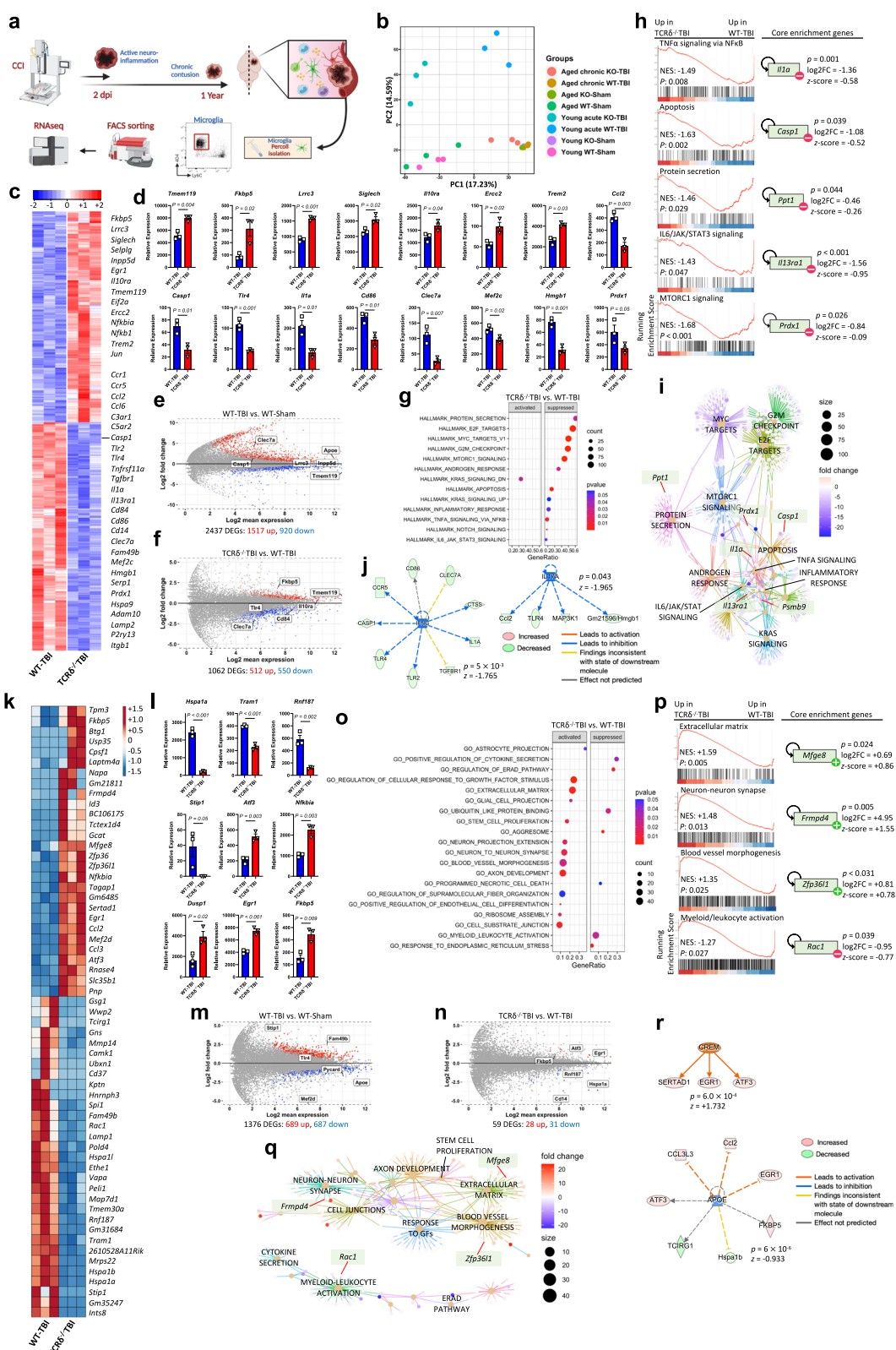

deposition[67]) were decreased whereas *Atf3* (involved in nerve regeneration[68]), *Nfkbia* (NF-κB inhibitor), *Dusp1* (M2-polarizing gene[69]) and *Egr1* (pro-growth factor[70]) were increased (Fig. 3k, l, n). GSEA analysis revealed enrichment in biological pathways involved in matrix deposition, growth, stem cell proliferation, neuron projection, axon development, synaptogenesis, cell junction assembly and reduction of cell death pathways in microglia isolated from aged chronic TCRδ⁻/⁻ TBI mice vs. age-matched WT-TBI mice (Fig. 3o, p). An interaction

cnetplot between the chronic pathways altered in TCRδ⁻/⁻ TBI microglia is shown in Fig. 3q. Of note, APOE and CAMP Responsive Element Modulator (CREM) were found to be major upstream regulators of chronic TBI pathogenesis (Fig. 3r).

We also investigated the effect of γδ T cells on the aging microglia. We found that microglia from aged TCRδ⁻/⁻ mice had only 84 DEGs vs.1879 DEGs in microglia from aged WT-TBI mice (Supplementary Fig. 3b–e) suggesting the involvement of γδ T cells in aging. For

**Fig. 3 | γδ T cells modulate microglia in acute and chronic TBI. a** Schematic of the experimental layout. Created with BioRender.com. **b** Principal component analysis of young (two days post-TBI) and aged (one year post-TBI), WT and TCRδ⁻/⁻, Sham and TBI, microglia transcriptomes. **c** Heatmap of the differentially expressed genes two days post-TBI. **d** Individual gene plots (WT-TBI *n* = 3, TCRδ⁻/⁻ TBI *n* = 3, where each *n* is a pool of two samples). **e, f** MA plot of WT-TBI vs. WT-Sham **e** and TCRδ⁻/⁻ TBI vs. WT-TBI **f** microglia genes. **g** GSEA of TCRδ⁻/⁻ TBI vs. WT-TBI pathways. **h** Individual TCRδ⁻/⁻ TBI vs. WT-TBI GSEA plots showing the normalized enrichment score and the corresponding FDR-adjusted *P* value for each pathway (left) and a top driving gene (right). **i** Cnetplot depicting the network of interacting genes among the enriched pathways. **j** IPA analysis showing the predicted regulators of the TCRδ⁻/⁻ TBI vs. WT-TBI differentially expressed microglia genes. **k** Heatmap of the differentially expressed genes one year post-TBI. **l** Individual gene

plots (WT-TBI *n* = 3, TCRδ⁻/⁻ TBI *n* = 3, where each *n* is a pool of two samples). **m** MA plot of WT-TBI vs. WT-Sham microglia genes. **n** MA plot of TCRδ⁻/⁻ TBI vs. WT-TBI microglia genes. **o** GSEA of TCRδ⁻/⁻ TBI vs. WT-TBI pathways. **p** Individual TCRδ⁻/⁻ TBI vs. WT-TBI GSEA plots and a top pathway gene (right). **q** Cnetplot depicting the network of interacting genes among the enriched pathways. **r** IPA analysis showing the predicted regulators of the aged TCRδ⁻/⁻ TBI vs. WT-TBI differentially expressed microglia genes. Biologically independent samples were used. Data shown as mean ± sem. *P* values calculated by two-tailed unpaired Student's *t*-test **d, l**, FDR-adjusted *P* values with DEseq2 using the Wald test for significance following fitting to a negative binomial linear model and the Benjamini-Hochberg procedure to control for false discoveries **e, f, m, n**, one-tailed *t*-test **g, h, o** and right-tailed Fisher's exact test **j, r**. Dpi days post-injury, WT wild-type, KO knock-out, NES normalized enrichment score, DEG differentially expressed gene.

---

instance, aging-associated genes such as *Clec7a*[61,71] and *Ccl2*[72] found to be upregulated in aged WT-TBI mice were not differentially expressed in aged TCRδ⁻/⁻ TBI mice.

Taken together, these findings indicate that brain-infiltrating γδ T cells following TBI induce a microglial pro-inflammatory phenotype that contributes to TBI pathogenesis.

## Opposing roles of Vγ1 and Vγ4 in neuroinflammation following TBI

γδ T cells in mice are classified according to the variable portion of the gamma chain of the TCR and their tissue distribution[40]. Vγ1 and Vγ4 γδ T cell subsets often show opposing effects in modulating immunity associated with diseases including autoimmune conditions[73,74] and tumors[75]. Moreover, these γδ T cell subsets can infiltrate the brain during CNS inflammation[28,29]. First, we performed a kinetic study in which we determined the frequencies of total γδ T cells and Vγ1 and Vγ4 subsets in the brain, blood, deep cervical lymph nodes (cLN) and spleen at 1, 3, 7, 28 days and 1 year after TBI. We found that γδ T cell frequencies increased 3 days after brain injury, peaked at 7 days and were still elevated 1 year post-injury (Supplementary Fig. 4a). Frequencies of γδ T cells in mice subjected to TBI were increased in the deep cLN, spleen and blood during the acute phase of the disease (1–28 days post injury; Supplementary Fig. 4a). There was a marked increase in Vγ1 subset in the brain as early as 1 day after TBI, which was maintained 3 days post-injury and decreased to levels of Sham mice by 7 days after TBI (Supplementary Fig. 4b). Conversely, Vγ4 subset, although less frequent than Vγ1 subset, increased in the brain of TBI mice 3 days after injury and continued to be elevated up to 1 year after injury (Supplementary Fig. 4b). Importantly, Vγ4 subset was completely absent in the brains of Sham mice (Supplementary Fig. 4b). The temporal profile of Vγ1 and Vγ4 γδ T cell subsets as well as a to-be-characterized Vγ1-Vγ4-cell population in the deep cervical lymph nodes, blood and spleen changed with time with a predominant expansion of the Vγ1 subset (Supplementary Fig. 4b).

Next, we investigated the effect of γδ T cell subsets in acute and chronic TBI by depleting Vγ1, Vγ4 or total γδ T cells 24 h before injury (Fig. 4a). γδ T cell depletion was validated by RT-qPCR (Supplementary Fig. 5a). In acute TBI, we found that Vγ4-depleted mice had less CD4+ (Fig. 4b) and CD8+ (Fig. 4c) infiltrating T cells and reduction of Ly6C^hi classical monocytes (Fig. 4d), CCR2 + CX3CR1+ monocytes (Fig. 4e) and MHCII expression on CD45^int microglia/macrophages (Fig. 4f) compared to PBS-treated mice. Vγ1-depleted mice showed increased infiltration of CD4+ (Fig. 4b) and CD8+ (Fig. 4c) T lymphocytes. Vγ4-depleted mice displayed a greater reduction of CNS inflammatory cytokines including *Il6*, *Il12*, *Il18* and *Tnf* whereas Vγ1-depleted mice had less regulatory cytokine expression including *Il10* and *Tgfb* (Fig. 4g; Supplementary Fig. 5b), which may account for the pro-inflammatory role of the Vγ4 γδ T cell subset in the CNS. We also investigated the integrity of the blood

brain barrier (BBB) in acute TBI and found that Vγ4-depleted mice had less BBB breakdown as shown by the normal levels of expression of occludin and ZO-1 (Fig. 4h).

We then investigated the effect of the two γδ T cell subsets on behavioral outcomes. We found that Vγ4-depleted mice had improved motor coordination (Fig. 4i) 2 weeks after TBI as well as increased locomotor activity (Fig. 4j) and reduced anxiety (Fig. 4k) 1 month after TBI. Vγ1-depleted mice exhibited impaired memory on the MWM (Fig. 4l–n) whereas Vγ4-depleted mice had improved memory 1 month after TBI (Fig. 4l–n). Furthermore, microglia isolated from Vγ4-depleted TBI mice had downregulation of inflammatory/activating genes including *Il1b*, *Cd36*, *CD74*, *Ccr2*, *Cfp*, *H2-Eb1*, *Hspa1a*, *Hspa1b*, *Ifitm1*, and *Chchd* compared to microglia isolated from Vγ1-depleted mice. Conversely, Vγ1-depleted mice had downregulation of homeostatic genes including *Tgfb* (Fig. 4o, p; Supplementary Data 2). GSEA revealed a downregulation of genes related to signaling pathways involved in the production of molecular mediators of immune response in microglia isolated from the Vγ4-depleted cohort (Fig. 4q). Thus, because depletion of the Vγ4 subset exhibited a similar neuroinflammatory profile to depletion of total γδ T cells (Figs. 1–3), our findings suggest that Vγ4 cells are the primary mediators of pathogenesis in acute TBI.

To further investigate the differential effects of γδ T cell subsets in acute and chronic TBI, we adoptively transferred Vγ1 or Vγ4 γδ T cells isolated from WT mice into TCRδ⁻/⁻ mice at the time of injury (Fig. 5a). Importantly, the TCRδ⁻/⁻ recipient mice were littermates of each other thus controlling for environmental factors. The cellular components adoptively transferred were validated by RT-qPCR (Supplementary Fig. 6a). In acute TBI, TCRδ⁻/⁻ mice that received WT-Vγ1 cells had reduced CD4+ (Fig. 5b) and CD8+ (Fig. 5c) infiltrating T cells with no difference in Ly6C^hi and Ly6C^int monocytes (Fig. 5d), but reduced CCR2 + CX3CR1+ monocytes (Fig. 5e) as compared to PBS-treated mice. Conversely, TCRδ⁻/⁻ mice that received WT-Vγ4 cells had increased CD4+ (Fig. 5b) infiltrating T cells, increased intra-parenchymal CCR2 + CX3CR1+ monocytes (Fig. 5e) and increased MHCII expression on CD45^int microglia/macrophages (Fig. 5f) as compared to PBS-treated mice. In contrast to Vγ4 γδ T cells, adoptive transfer of Vγ1 γδ T cells reduced acute CNS neuroinflammatory cytokines, increased *Tgfb1* expression (Fig. 5g; Supplementary Fig. 6b) and led to less breakdown of the BBB (Fig. 5h). We then investigated whether adoptive transfer of the Vγ1 γδ T cell subset would have a therapeutic effect on the long-term neurocognitive deficits after TBI. For this, we adoptively transferred WT Vγ1 or Vγ4 γδ T cells into WT recipients 24 h and at 1 week after injury. We found that Vγ1 γδ T cells improved motor coordination (Fig. 5i) and reduced anxiety-like behavior (Fig. 5k) 1 month after TBI. The MWM test showed improved memory function (Fig. 5l) as Vγ1 recipients spent more time in the target quadrant (Fig. 5m) and crossed the target quadrant more frequently (Fig. 5n). In contrast, there was no change in neurocognitive deficits in Vγ4 recipients (Fig. 5i–n).

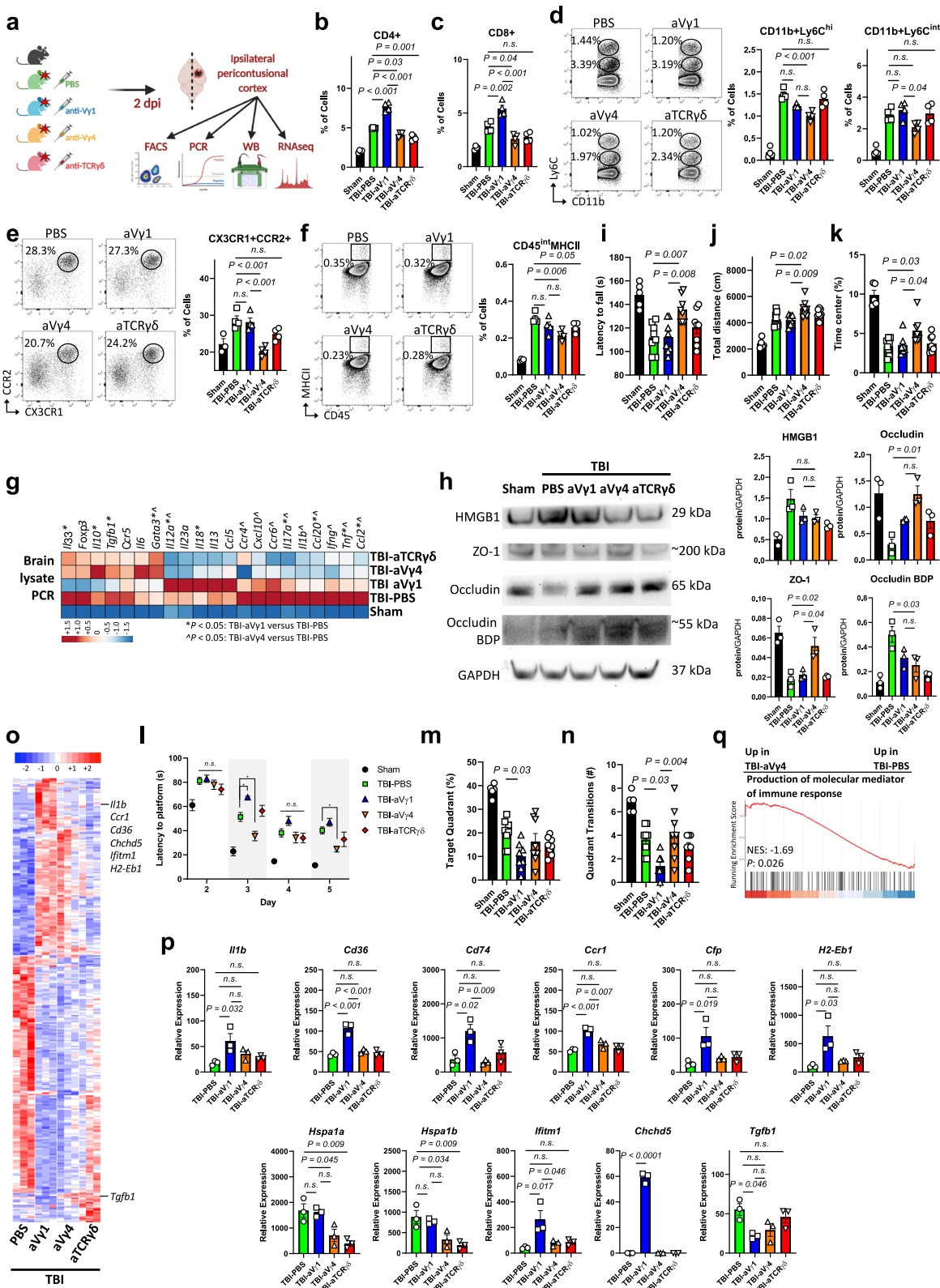

The improved behavioral outcomes in TCRδ[−/−] mice subjected to TBI who received Vγ1 γδ T cells correlated with their microglia phenotype. Microglia transcriptome analysis of mice that received Vγ1 γδ T cells revealed downregulation of microglia stress genes including *Hspa8*, *Hsph1*, *Ran* and *Mapk14* whereas homeostatic and regulatory genes such as *Siglech*, *Adgrg1*, *Itgam*, *Entpd1* and *Cmtm6* were increased (Fig. 5o, p; Supplementary Data 3).

Furthermore, GSEA analysis of the microglia transcriptome showed a downregulation of genes related to innate immune responses (Fig. 5q). Taken together, these findings demonstrate that γδ T cell subsets play opposing roles in TBI pathogenesis. While Vγ4 γδ T cells are detrimental and promote neuroinflammation, Vγ1 γδ T cells are beneficial and protect from TBI-induced neuroinflammation.

**Fig. 4 | Depletion of Vγ4 cells ameliorates TBI pathogenesis. a** Schematic of the experimental layout. Created with BioRender.com. **b–f** Flow cytometry of brain CD3 + CD4+ **b**, CD3 + CD8+ **c**, CD11b+Ly6C$^{hi}$ and CD11b+Ly6C$^{int}$ **d**, CD11b +CD45$^{hi}$CX3CR1 + CCR2+ **e**, and CD45$^{int}$MHCII+ cells **f** (Sham $n$ = 4, TBI-PBS $n$ = 4, TBI-aVγ1 $n$ = 4, TBI-aVγ4 $n$ = 4, TBI-aTCRγδ $n$ = 4). **g** RT-qPCR of cytokines and factors; (*) $P$ < 0.05 for TBI-aVγ1 versus TBI-PBS, (^) $P$ < 0.05 for TBI-aVγ4 versus TBI-PBS (see Supplementary Fig. 5b). **h** Western blot (left) and quantification (right) of the 29 kDa HMGB1, 200 kDa ZO-1, 65 kDa occludin, ~55 kDA occludin breakdown product and 37 kDa GAPDH (Sham $n$ = 3, TBI-PBS $n$ = 3, TBI-aVγ1 $n$ = 3, TBI-aVγ4 $n$ = 3, TBI-aTCRγδ $n$ = 3). **i** Rotarod test at two weeks post-TBI (Sham $n$ = 5, TBI-PBS $n$ = 8, TBI-aVγ1 $n$ = 8, TBI-aVγ4 $n$ = 8, TBI-aTCRγδ $n$ = 8). **j, k** Open field test one month post-TBI (Sham $n$ = 5, TBI-PBS $n$ = 8, TBI-aVγ1 $n$ = 8, TBI-aVγ4 $n$ = 8, TBI-aTCRγδ $n$ = 8). **l–n** Morris water maze test one month post-TBI (Sham $n$ = 5, TBI-PBS $n$ = 8, TBI-aVγ1 $n$ = 8, TBI-aVγ4 $n$ = 8, TBI-aTCRγδ $n$ = 8). (*) $P$ < 0.05. **o** Heatmap of the differentially expressed microglia genes two days post-injury. **p** Individual gene plots (TBI-PBS $n$ = 3, TBI-aVγ1 $n$ = 3, TBI-aVγ4 $n$ = 3, TBI-aTCRγδ $n$ = 3, where each $n$ is a pool of two samples). **q** GSEA plot of one of the top TBI-aVγ4 versus TBI-PBS gene ontology pathways. *$P$ < 0.05. Biologically independent samples were used. Data shown as mean ± sem. $P$ values calculated by one-way ANOVA with Bonferroni's multiple comparison test **b–f, h–k, m, n, p**, two-way repeated measures ANOVA with Bonferroni's multiple comparison test **l** and one-tailed $t$-test **q**. Dpi, days post-injury; aVγ1, anti-Vγ1 depleting antibody; aVγ4, anti-Vγ4 depleting antibody; aTCRγδ, anti-TCRγδ depleting antibody; BDP, breakdown product; NES, normalized enrichment score. Source data are provided as a Source Data file.

## Vγ1 γδ T cells produce TGF-β to reduce microglia activation and ameliorate neuroinflammation

As shown above, Vγ1-depleted TBI mice had reduced brain expression of *Tgfb* and microglial *Tgfb1* (Fig. 4g, o, p), and TCRδ$^{−/−}$ mice that received WT-Vγ1 cells had increased expression of *Tgfb* (Fig. 5g). Although Vγ1 expressing γδ T cells can produce proinflammatory cytokines including IFN-γ and IL-17A[76], we hypothesized that the beneficial effects of Vγ1 cell transfer on TBI neuroinflammation were related to anti-inflammatory or modulatory cytokines that these cells can secrete. Consistent with this, we have previously found that γδ T cells from both mice and humans can express the membrane-bound TGF-β (termed latency-associated peptide (LAP)), which confers these cells the ability to secrete TGF-β and dampen inflammation[42].

To investigate whether the beneficial effects of Vγ1 cell transfer on TBI were dependent on TGF-β produced by Vγ1 γδ T cells, we first performed flow cytometric analysis of splenocytes from naïve mice to measure LAP expression on γδ T cell subsets. We found that Vγ1 γδ T cells expressed more LAP compared to Vγ4 (Supplementary Fig. 6c). Of note, we found that Vγ1 γδ T cells isolated from the deep cervical lymph nodes and spleen reactively produced more IFN-γ at 3 days after TBI, however, Vγ1 cells were found to secrete little to no IL-17 (Supplementary Fig. 6d, e). We then adoptively transferred total WT-Vγ1 cells or WT-Vγ1LAP$^{lo}$ cells into TCRδ$^{−/−}$ recipients at the time of injury to investigate the role of LAP-expressing Vγ1 subset in acute TBI (Fig. 6a; Supplementary Fig. 7a). We found that TCRδ$^{−/−}$ mice that received total WT-Vγ1 cells had reduced infiltrating CD4+ (Fig. 6b) and CD8+ (Fig. 6c) T cells as well as reduced Ly6C$^{hi}$ and Ly6C$^{int}$ monocytes (Fig. 6d) with no difference in CX3CR1 + CCR2+ monocytes (Fig. 6e) but had reduced MHCII expression on CD45$^{int}$ microglia/macrophages (Fig. 6f) as compared to mice that received the WT-Vγ1LAP$^{lo}$ cell population. Moreover, mice that received total WT-Vγ1 cells had a greater reduction of CNS inflammatory cytokines including *Il18*, *Il1b* and *Tnf* (Fig. 6g; Supplementary Fig. 7b). Microglia isolated from recipients of WT-Vγ1LAP$^{lo}$ cells exhibited an activated transcriptional profile with upregulation of genes including *Clec7a*, *Il1a*, *Il16*, *Cd86*, *Hspa1b*, and *Lag3* along with downregulation of homeostatic genes including *Fam49b*, *Socs4*, and *Irf2bpl* (Fig. 6h, i; Supplementary Data 4). Consistent with this, microglia from WT-Vγ1LAP$^{lo}$ recipient mice had enrichment in biological pathways involved in chemokine production and lymphocyte activation (Fig. 6j). To further investigate the TGF-β-dependent Vγ1-microglia crosstalk, we employed an ex-vivo transwell co-culture system in which Vγ1 γδ T cells isolated from WT-Sham mice were plated in the upper chamber with or without anti-TGF-β neutralizing antibody and were co-cultured with microglia isolated from WT-TBI mice plated in the lower chamber (Fig. 6k). RNA-seq of microglia co-cultured with Vγ1 γδ T cells and anti-TGF-β had upregulation of microglia activation genes including *Ccl5*, *Cx3cr1*, *Nfkb2*, *Tnfrsf26* and *Rock1* (Fig. 6l, m; Supplementary Data 4). Thus, Vγ1 γδ T cells ameliorate acute TBI immunopathology by secreting TGF-β.

## Vγ4 γδ T cells produce IL-17 and IFN-γ to activate microglia and induce neuroinflammation

Vγ4 γδ T cells are known to produce IL-17 and IFN-γ[75,77] and both cytokines negatively affect the pathogenesis of CNS injury[28,29]. To investigate IL-17 and IFN-γ expression in TBI, we measured the frequencies of IL-17 and IFN-γ in Vγ4 γδ T cells in the deep cLN and spleen at 1, 3, 7, 28 days and 1 year after TBI. We were not able to investigate cytokine expression in Vγ1 and Vγ4 subsets in the brain due to the low frequency of these cells (Supplementary Fig. 4b). We found that IL-17-but not IFN-γ-producing Vγ4 γδ T cells were increased in the cLN from TBI mice at all time-points analyzed whereas both IL-17 and IFN-γ-producing Vγ4 γδ T cells were increased in the spleen from TBI mice particularly at later time-points (28 days and 1 year post-injury; Supplementary Fig. 8a).

To further investigate the role of IFN-γ and IL-17 produced by Vγ4 γδ T cells in TBI, we adoptively transferred total WT-Vγ4 cells isolated from WT, IFN-γ$^{−/−}$ or IL17af$^{−/−}$ mice into TCRδ$^{−/−}$ recipients at the time of injury (Fig. 7a). We found that TCRδ$^{−/−}$ mice that received IFN-γ$^{−/−}$Vγ4 cells had reduced infiltrating CD8+ (but not CD4+) T cells (Fig. 7b, c), reduced Ly6C$^{hi}$ classical monocytes (Fig. 7d, e), and less MHCII expression on CD45$^{int}$ microglia/macrophages (Fig. 7f) compared to mice that received WT-Vγ4 cells. While both cytokines contribute to acute post-TBI neuroinflammation, IFN-γ$^{−/−}$Vγ4 recipients displayed greater reduction of CNS inflammatory cytokines including *Tnf*, *Ifng*, and *Il1b* as compared with IL17af$^{−/−}$Vγ4 recipients, which upregulated *Tgfb* (Fig. 7g; Supplementary Fig. 8b). Given that Vγ4 γδ T cells led to microglial activation (Figs. 4 and 5), we asked whether IL-17 and/or IFN-γ played a role in the Vγ4-microglia crosstalk in acute TBI. Whereas microglia isolated from TCRδ$^{−/−}$ recipients of both IL17af$^{−/−}$Vγ4 and IFNγ$^{−/−}$Vγ4 cells displayed a homeostatic transcriptional profile versus recipients of total Vγ4 γδ T cells, IL17af$^{−/−}$Vγ4 recipients had the most prominent homeostatic profile (Fig. 7h, i), likely due to the abundance of TGF-β in the absence of IL-17 (Fig. 7g). Adoptive transfer of IL17af$^{−/−}$Vγ4 cells led to further downregulation of genes including *Sod1*, *Cd72*, Spp1, *Ccl12*, *Cstb*, *Bad* and *Hspa1a* along with increased expression of *Cx3cr1* and *Lrp1* (Fig. 7h, i; Supplementary Data 5). Accordingly, in microglia isolated from IL17af$^{−/−}$Vγ4 recipients, there was a downregulation of genes related to biological pathways involved in response to reactive oxygen species (Fig. 7j). To further investigate the role of IFN-γ in Vγ4-microglia crosstalk, we designed a GFP-shRNA lentivirus that targets IFN-γR1 specifically on CD11b+ cells (Fig. 7k). We validated the virus using flow cytometry (Fig. 7l) and RT-qPCR (Fig. 7m). Microglia isolated two days after TBI from animals that were intraventricularly injected with the shRNA-IFN-γR1 at the time of injury exhibited a more homeostatic microglial profile in which *Irf8*, *Stat3*, *Junb* and *Egr1* were downregulated and *Siglech* and *Csf1r* were upregulated compared to microglia isolated from animals injected with the corresponding scrambled lentivirus (Fig. 7n, o; Supplementary Data 5). GSEA analysis revealed that IFN-γR1-deficient microglia had a downregulation of genes related to cytokine-mediated signaling (Fig. 7p). To investigate the role of IL-17

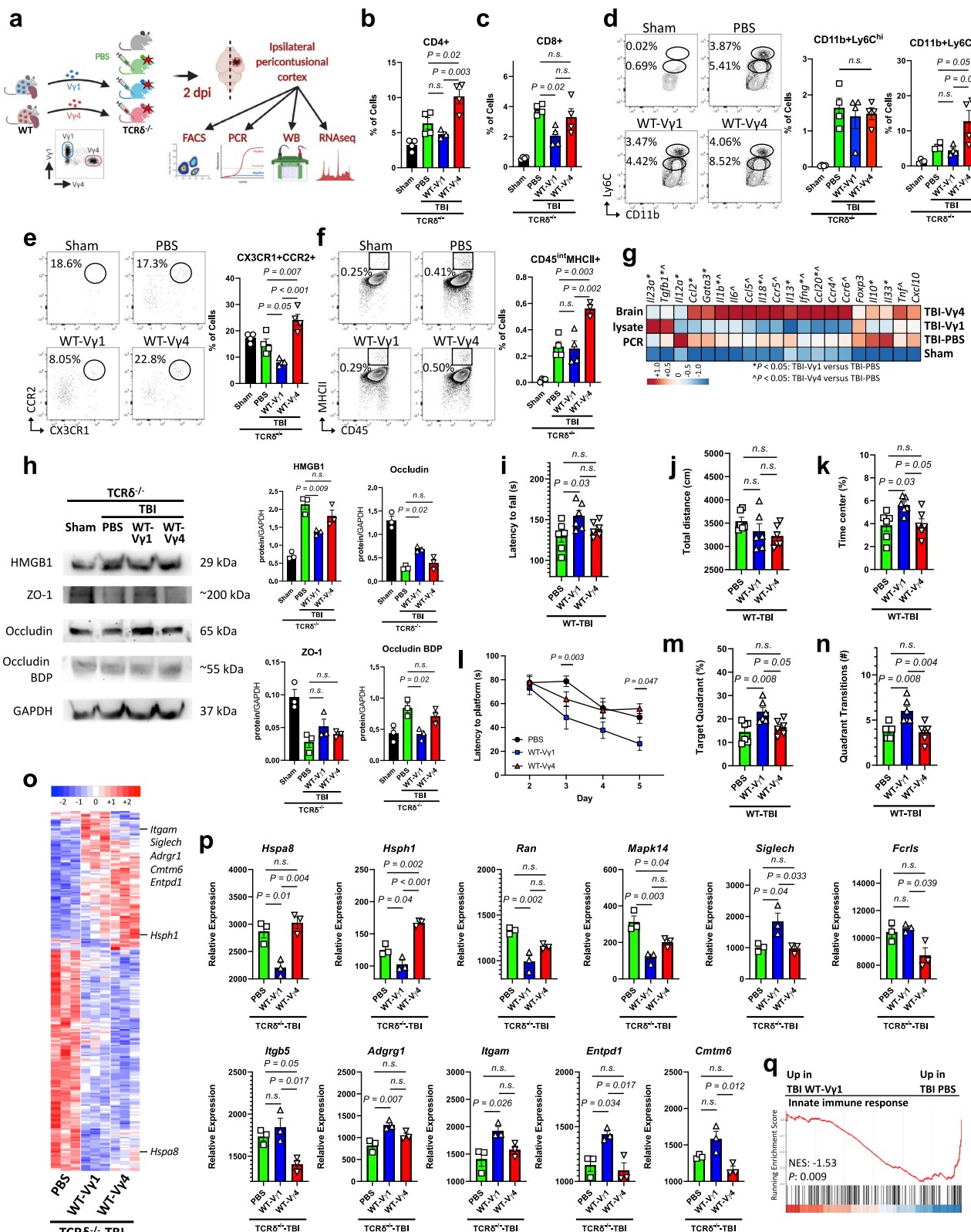

in Vγ4-microglia crosstalk, we used the ex-vivo transwell co-culture system described above (Figs. 6k and 7q). RNA-seq of microglia co-cultured with Vγ4 γδ T cells and anti-IL-17A neutralizing antibody revealed downregulation of microglia activation genes including *Casp2*, *Traf1*, *Cd69* and *Il12rb2* (Fig. 7r, s; Supplementary Data 5). Taken together, this is consistent with our finding that γδ T cell-secreted IFN-γ and IL-17A act as upstream regulators of microglia

(Fig. 3j). Thus, Vγ4 γδ T cells contribute to TBI pathogenesis by activating microglial cells via IFN-γ and IL-17.

## Discussion

The role of the peripheral immune response in TBI remains poorly understood. Neuroglia-lymphocyte crosstalk has been identified as a critical mediator of TBI pathogenesis[78]. Here, we report that γδ T

**Fig. 5 | Adoptive transfer of Vγ1 cells ameliorates TBI pathogenesis. a** Schematic of the experimental layout. Created with BioRender.com. **b**–**f** Flow cytometry of brain CD3 + CD4+ **b**, CD3 + CD8+ **c**, CD11b+Ly6C$^{hi}$ and CD11b+Ly6C$^{int}$ **d**, CD11b+CD45$^{hi}$CX3CR1 + CCR2+ **e**, and CD45$^{int}$MHCII+ cells **f** (Sham $n = 4$, PBS $n = 4$, WT-Vγ1 $n = 4$, WT-Vγ4 $n = 4$). **g** RT-qPCR of cytokines and factors; (*) $P < 0.05$ for TBI-Vγ1 versus TBI-PBS, (^) $P < 0.05$ for TBI-Vγ4 versus TBI-PBS (see Supplementary Fig. 6b). **h** Western blot (left) and quantification (right) of the 29 kDa HMGB1, 200 kDa ZO-1, 65 kDa occludin, -55 kDA occludin breakdown product and 37 kDa GAPDH (Sham $n = 3$, PBS $n = 3$, WT-Vγ1 $n = 3$, WT-Vγ4 $n = 3$). **i** Rotarod test at two weeks post-TBI (PBS $n = 6$, WT-Vγ1 $n = 6$, WT-Vγ4 $n = 6$). **j**, **k** Open field test one month post-TBI **j**, **k** (PBS $n = 6$, WT-Vγ1 $n = 6$, WT-Vγ4 $n = 6$). **l**–**n** Morris water maze test one month post-TBI (PBS $n = 6$, WT-Vγ1 $n = 6$, WT-Vγ4 $n = 6$). **o** Heatmap of the differentially expressed microglia genes among PBS, WT-Vγ1 and WT-Vγ4 groups, two days after injury. **p** Individual plots of genes of interest (PBS $n = 3$, WT-Vγ1 $n = 3$, WT-Vγ4 $n = 3$, where each $n$ is a pool of two samples). **q** GSEA plot of one of the top gene ontology pathways in the WT-Vγ1 versus PBS group. Biologically independent samples were used. Data shown as mean ± sem. $P$ values calculated by one-way ANOVA with Bonferroni's multiple comparison test (**b**–**f**, **h**–**k**, **m**, **n**, **p**), two-way repeated measures ANOVA with Bonferroni's multiple comparison test **l** and one-tailed $t$-test **q**. Dpi days post-injury, BDP breakdown product, NES normalized enrichment score. Source data are provided as a Source Data file.

cells modulate acute neuroinflammation and chronic functional deficits following TBI. We also characterized TBI-associated microglia signatures and their modulation by γδ T cells in acute and chronic TBI. We found that Vγ4 expressing γδ T cells promote TBI by activating microglial cells via IFN-γ and IL-17 whereas Vγ1 expressing γδ T cells ameliorate TBI by favoring a homeostatic microglial signature via TGF-β. Furthermore, more attention has been recently directed towards understanding the role of the gut-brain axis in TBI. In our previous studies[55], we found that transferring microbiota from TCRδ$^{-/-}$ mice worsened intestinal inflammation, impaired oral tolerance, increased IL-17 production and decreased Tregs. In this study, we find the opposite effect in TCRδ$^{-/-}$ mice, suggesting that the genetic deletion, rather than microbiome changes, account for the protective effect. Consistent with the inflammatory role of γδ T cells in brain injury, Benakis et al. found that antibiotics altered the gut microbiota and decreased IL-17 producing γδ T cells in an animal mode of ischemic stroke[79]. While both the Benakis study and the present study find that reducing γδ T cells is protective in brain injury, here we used a genetic approach rather than a microbiome approach.

TBI is associated with an increased risk of developing neurological disorders including chronic traumatic encephalopathy[8], Alzheimer's disease[5], Parkinson's disease[80], amyotrophic lateral sclerosis[7], and psychiatric disorders[81–83], all of which involve microglial dysfunction and impaired clearance of danger-associated molecular patterns (DAMP). In our study, we found that γδ T cells infiltrated the CNS following TBI and the crosstalk between γδ T cells and microglia modulated post-traumatic neuroinflammation. Consequently, accumulation of neurotoxic DAMPs such as HMGB1, BBB breakdown, APP and p-Tau deposition as well as chronic behavioral deficits were observed. Microglia isolated from TCRδ$^{-/-}$ TBI mice exhibited ATF-3-driven pro-metabolic and pro-regenerative molecular signatures[84,85]. Compared to WT-TBI, TCRδ$^{-/-}$ TBI microglia exhibited upregulation of two pro-regenerative genes, *Atf3*[68] and *Egr1*[70]. Aging-associated genes such as microglial *Clec7a*[61,71] and *Ccl2*[72], which also represent a neurodegenerative microglial (MGnD) phenotype[61], were upregulated in aged WT-TBI, but not in aged TCRδ$^{-/-}$ TBI mice. Furthermore, APOE, which was reported to be a major regulator of MGnD microglia[61], was predicted to be downregulated in aged TCRδ$^{-/-}$ TBI mice. Thus, γδ T cell/microglia interactions are important drivers of neurodegeneration in TBI[86].

Although γδ T cells comprise the majority of immune cells in niches associated with epithelial surfaces such as the intestine, only 1–2% of γδ T cells are present in secondary lymphoid tissues[87]. Upon BBB disruption after TBI, γδ T cell expand in the secondary lymphoid organs and migrate to the CNS via the blood[88] and/or lymphatics[38,89,90]. We found expansion of total γδ T cells in the deep cervical lymph nodes, spleen, blood, and brain. The Vγ1 subset predominated in all sites investigated. In the brain, Vγ1 γδ T cells infiltrated as quickly as 1 day after injury, remained elevated at 3 days and decreased at 7 days post-injury. Conversely, Vγ4 γδ T cells increased at 3 days after TBI and remained elevated up to 1 year post injury concurrent with a decrease of Vγ4 cells in the deep cervical lymph nodes. We found significant decrease in a yet undefined double-negative Vγ1-Vγ4- cell population in the brain at 3 days after TBI which requires further characterization. Because depletion of Vγ1 γδ T cells worsened TBI-induced neuroinflammation, activated microglial cells, and contributed to the functional impairment following brain injury, and because transfer of Vγ1 cells ameliorated inflammation, reduced microglial activation, and improved cognition following TBI, we concluded that Vγ1 γδ T cells play a beneficial role in TBI. Although Vγ1 γδ T cells can produce pro-inflammatory cytokines including IFN-γ and IL-17[76], which we found to be involved in Vγ4 cell-mediated neuroinflammation post-TBI, it was possible that a subset of regulatory Vγ1 cells also infiltrated the brain and regulated inflammation. This hypothesis was based on our previous study in which we found that a subset of regulatory γδ T cells that express LAP, a membrane-bound TGF-β, exist in both mice and humans[42]. Consistent with this, we found that neuroprotection conferred by Vγ1 γδ T cells following TBI was abrogated in the absence of Vγ1+LAP$^{hi}$ cells in vivo and after TGF-β neutralization in vitro, indicating that LAP-expressing Vγ1 γδ T cells were responsible for the beneficial effects of Vγ1 γδ T cells in TBI. The anti-inflammatory activity of Vγ1+LAP$^{hi}$ cell was likely a result of a TGF-β dependent suppression of microglia activation, increase of amyloid-beta clearance by microglia, and reduction of neuronal cell death, which are known functions of TGF-β in brain inflammation[91–94].

Despite the lower frequency of Vγ4 γδ T cells in the brain of mice with TBI compared with Vγ1 γδ T cells, Vγ4 cells played a detrimental role in TBI-induced neuroinflammation by producing IFN-γ and IL-17. We showed that lentiviral knockdown of IFN-γR1 on microglial (CD11b+) cells, neutralization of IL-17 in an ex-vivo transwell co-culture system, and adoptive transfer of Vγ4 cells from WT IFNγ$^{-/-}$ or IL17af$^{-/-}$ into TCRδ$^{-/-}$ recipients ameliorated neuroinflammation, reduced microglial activation and improved cognition. Consistent with this, in a mouse model of stroke, γδ T cells, but not CD4 + T cells, were major sources of IL-17 within the brain and IL-17$^{-/-}$ mice had reduced infarct volume at day 4 after stroke[28]. It has been shown that accumulation of meningeal IL-17+ γδ T cells was associated with increased size of the acute infarct[79] whereas increased levels of IL-17A at 14 days after stroke promoted neural recovery[95] suggesting a paradoxical and time-sensitive effect of IL-17 in brain injuries. In a model of spinal cord injury, TCRδ$^{-/-}$ mice had improved functional recovery[29] and the Vγ4 subset was found to be the major mediator of neuroinflammation via IFN-γ whereby treatment with anti-Vγ4 antibody improved motor function[29].

Our study reveals previously unknown roles of Vγ1 and Vγ4 γδ T cell subsets in experimental TBI (Supplementary Fig. 10a, b). However, it is important to mention some few limitations. First, the use of the genetically modified TCRδ$^{-/-}$ model may interfere with the maturation of other lymphoid and myeloid cells, and contribute to the observed effects seen in Figs. 1–3. Moreover, it may be argued that the lack of γδ T cells in all tissues from birth as well as the baseline behavioral differences complicates our interpretation of the behavioral phenotype after TBI. Second, we found some differences in behavioral and immunologic effects between TCRδ$^{-/-}$ mice and mice treated with anti-TCRγδ depleting mAb. This could be related to the fact that the dose of

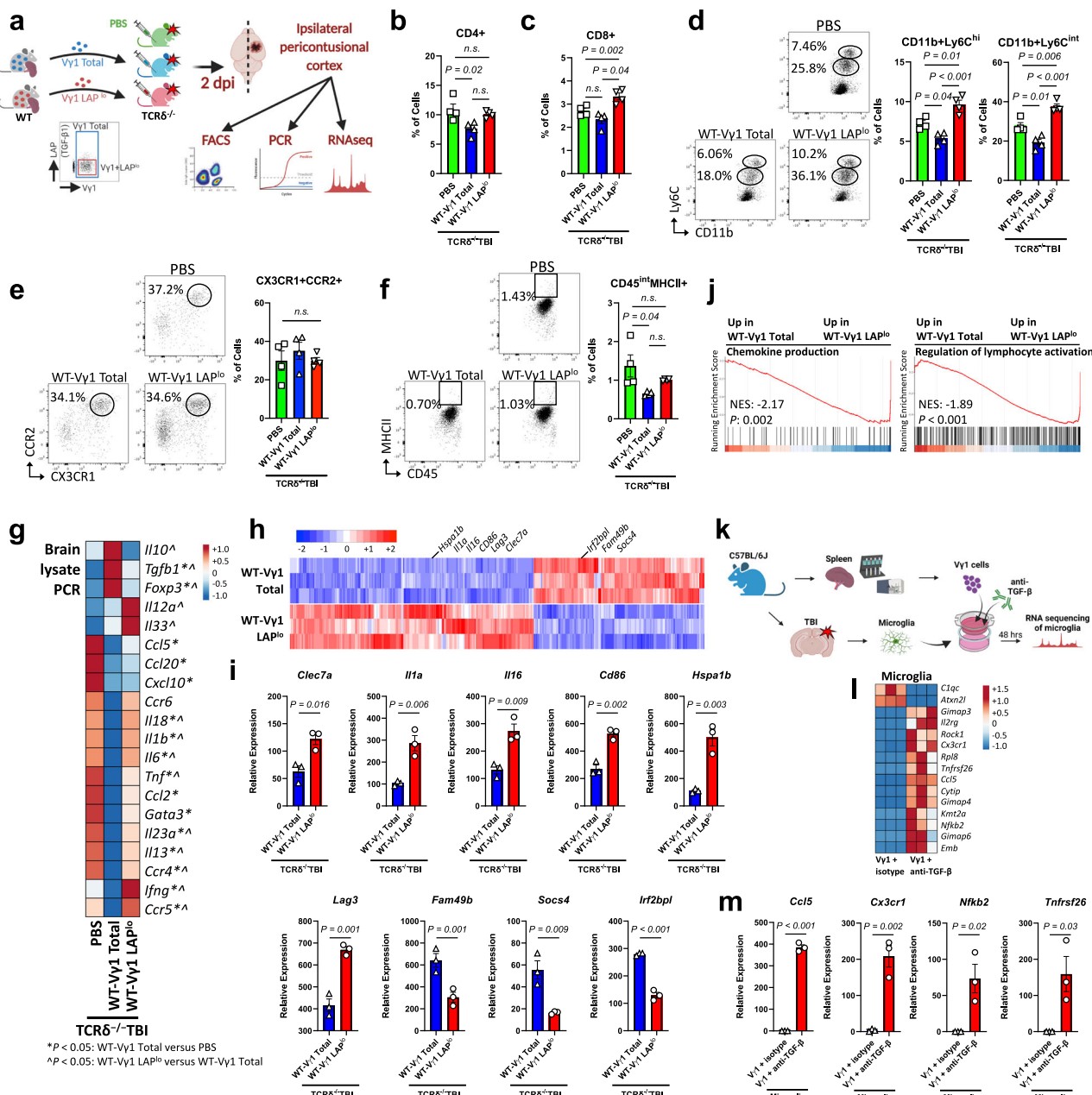

**Fig. 6 | Vγ1 cells ameliorate neuroinflammation after TBI via TGF-β. a** Schematic of the experimental layout. Created with BioRender.com. **b–f** Flow cytometric quantification of brain CD3 + CD4 + T cells **b**, CD3 + CD8 + T cells **c**, CD11b+Ly6Chi and CD11b+Ly6Cint **d**, CD11b+CD45hiCX3CR1 + CCR2+ **e** and CD45intMHCII+ myeloid cells **f** (PBS n = 4, WT-Vγ1 total n = 4, WT-Vγ1LAPlo n = 4). **g** Quantitative RT-qPCR of cytokines and factors measured from brain lysate at two days after TBI where (*) indicates statistical significance P < 0.05 for WT-Vγ1 versus PBS, (^) indicates statistical significance P < 0.05 for WT-Vγ1LAPlo versus WT-Vγ1 (see Supplementary Fig. 7b for quantification). **h** Heatmap of the differentially expressed microglia genes between the WT-Vγ1 and WT-Vγ1LAPlo groups, two days after injury. **i** Individual plots of genes of interest (WT-Vγ1 n = 3, WT-Vγ1LAPlo n = 3, where each n

is a pool of two samples). **j** GSEA plots of two of the top gene ontology pathways in the WT-Vγ1LAPlo versus WT-Vγ1 group. **k** Schematic of the experimental layout. Microglia isolated from wild-type TBI mice were co-cultured in a transwell system with Vγ1 cells, with a TGF-β neutralizing antibody or its isotype for 48 h, after which microglia were isolated for RNAseq. Created with BioRender.com. **l** Heatmap of the differentially expressed microglia genes. **m** Individual plots of genes of interest (Vγ1 + isotype n = 3, Vγ1 + anti-TGF-β n = 3, where each n is a pool of two samples). Biologically independent samples were used. Data shown as mean ± s.e.m. P values calculated by one-way ANOVA with Bonferroni's multiple comparison test **b–f**, two-tailed unpaired Student's t-test **i, m** and one-tailed t-test **j**. Dpi days post-injury, NES normalized enrichment score. Source data are provided as a Source Data file.

the depleting antibodies we used was based on what has been reported in literature rather than an in-house gradient-guided optimal concentration selection. Moreover, it has been shown that the anti-TCRγδ antibody clone UC7-13D5 may cause TCR internalization[96]. Thus, it is possible that although the frequency of γδ T cells is reduced in mice treated with anti-TCRγδ mAb, these cells may still be functional at some extent. Third, while Vγ4 recipients demonstrated a worsened neuroinflammatory milieu, the lack of any phenotypic effect may be

attributed to the chosen time point or the number of adoptively transferred cells. Fourth, the use of the designed lentivirus may inhibit the expression of IFN-γR1 on all brain CD11b-expressing cells and not only microglia. Lastly, littermate controls were not consistently used throughout the study and the behavior results observed in Figs. 1–3 could be partially attributed to particular gut commensals.

In summary, we demonstrated that γδ T cells play a critical role in the pathogenesis of acute and chronic TBI. Whereas Vγ4 γδ T cells

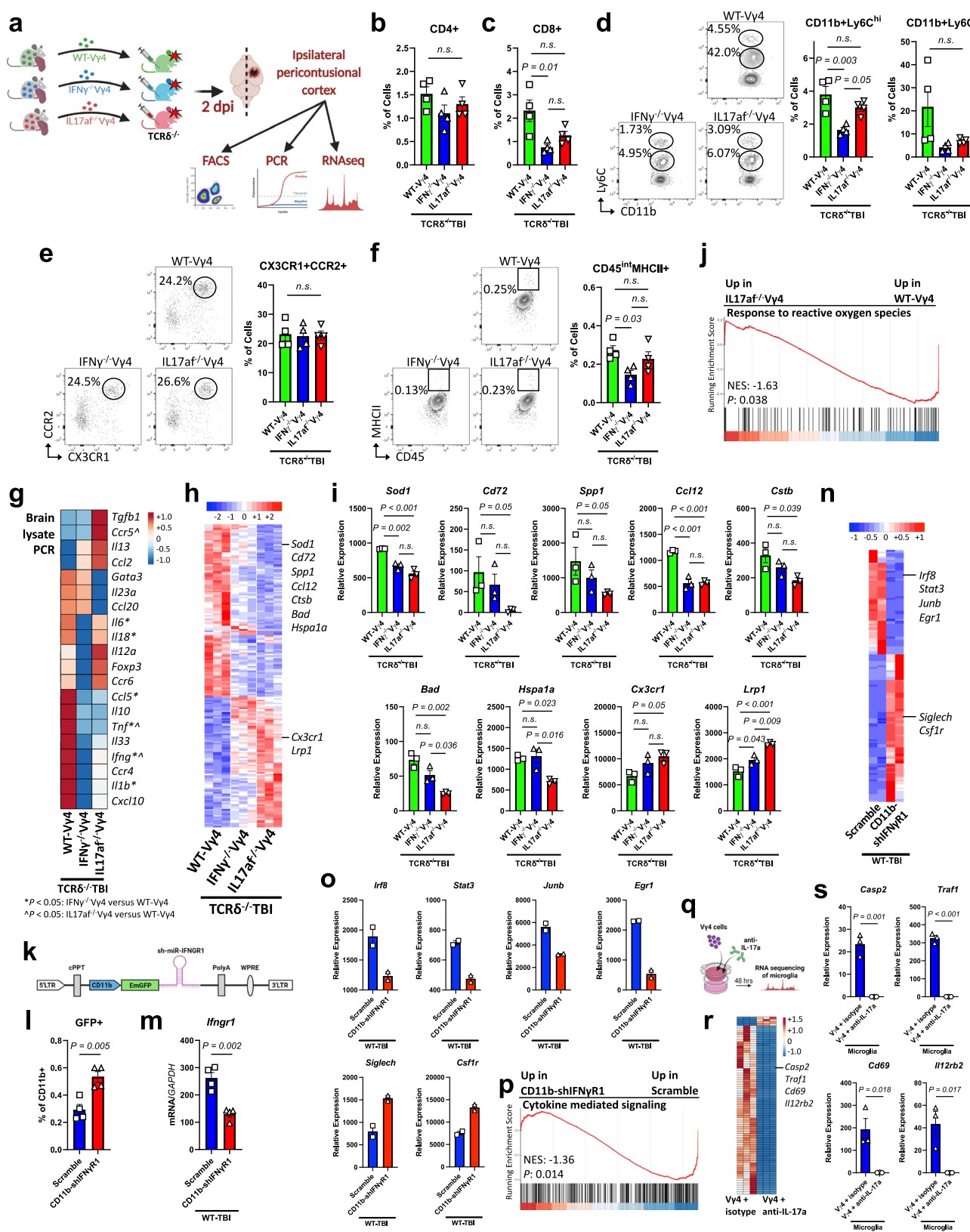

promote inflammation by activating microglial cells, Vγ1 γδ T cells ameliorate inflammation by promoting a homeostatic microglial signature. Thus, targeted modulation of Vγ4 or Vγ1 γδ T cells is a potential therapeutic approach to control neuroinflammation and mitigate neurological deficits after TBI.

# Methods

## Animals

Male and female mice were used in this study. In each figure, male mice were used unless otherwise specified in the legend. C57BL/6 J wild-type (WT) (#000664), TCRδ$^{-/-}$ (#002120), IFNγ$^{-/-}$ (#002287), and IL17af$^{-/-}$

**Fig. 7 | Vγ4 cells drive neuroinflammation after TBI via IL-17 and IFN-γ.**
**a** Schematic of the experimental layout. Created with BioRender.com. **b–f** Flow
cytometry of brain CD3 + CD4+ **b**, CD3 + CD8+ **c**, CD11b+Ly6C[hi] and CD11b+Ly6C[int]
**d**, CD11b+CD45[hi]CX3CR1 + CCR2+ **e**, and CD45[int]MHCII+ cells **f** (WT-Vγ4 $n = 4$,
IFNγ[-/-]Vγ4 $n = 4$, IL17af[-/-]Vγ4 $n = 4$). **g** RT-qPCR of cytokines and factors; (*) $P < 0.05$
for IFNγ[-/-]Vγ4 versus WT-Vγ4, (^) $P < 0.05$ for IL17af[-/-]Vγ4 versus WT-Vγ4 (see
Supplementary Fig. 8b). **h** Heatmap of the differentially expressed genes two days
post-injury. **i** Individual gene plots (WT-Vγ4 $n = 3$, IFNγ[-/-]Vγ4 $n = 3$, IL17af[-/-]Vγ4 $n = 3$,
where each $n$ is a pool of two samples). **j** GSEA plot a top gene ontology pathway in
the IL17af[-/-]Vγ4 versus WT-Vγ4 group. **k** Schematic of the designed IFN-γR1-
targeting lentiviral vector. Created with BioRender.com. **l** Flow cytometric quan-
tification of CD11b+ cells expressing GFP isolated from naïve brains intraven-
tricularly injected with the CD11b-shIFNγR1 lentivirus or the scramble (Scramble
$n = 4$, CD11b-shIFNγR1 $n = 4$). **m** Quantitative RT-qPCR of the level of expression of

*Ifngr1* mRNA in 4D4+ microglia cells isolated from TBI brains intraventricularly
injected with the CD11b-shIFNγR1 lentivirus or the scramble (Scramble $n = 4$, CD11b-
shIFNγR1 $n = 4$). **n** Heatmap of the differentially expressed microglia genes two days
post-injury. **o** Individual gene plots (Scramble $n = 2$, CD11b-shIFNγR1 $n = 2$, where
each $n$ is a pool of two samples). **p** GSEA plot of a top gene ontology pathway in the
CD11b-shIFNγR1 versus Scramble group. **q** Schematic of the experimental layout.
Created with BioRender.com. **r** Heatmap of the differentially expressed microglia
genes. **s** Individual gene plots (Vγ4 + isotype $n = 3$, Vγ4 + anti-IL-17a $n = 3$, where
each $n$ is a pool of two samples). Biologically independent samples were used. Data
shown as mean ± sem. $P$ values calculated by one-way ANOVA with Bonferroni's
multiple comparison test **b–f**, **i**, one-tailed $t$-test **j**, **p** and two-tailed unpaired Stu-
dent's $t$-test **l**, **m**, **s**. Dpi days post-injury, NES normalized enrichment score, GFP
green fluorescent protein. Source data are provided as a Source Data file.

(#034140) mice, 8-12 weeks old, were purchased from the Jackson
Laboratory. Animals were housed in a conventional specific pathogen-
free facility at the Building for Transformative Medicine, Brigham and
Women's Hospital, under standard 12-h light/dark cycle conditions.
TCRδ[-/-] mice were bred to generate TCRδ[-/-] litters. Carbon dioxide
administered for 2 min from a compressed gas tank followed by cer-
vical dislocation was used as the method of euthanasia. All experi-
ments were reviewed and overseen by the institutional animal care and
use committee at Brigham and Women's Hospital in accordance with
NIH guidelines for the humane treatment of animals (protocol number
2016N000230).

## Controlled cortical impact
An electromagnetic open-skull controlled cortical impact (CCI) model
of TBI was employed as previously described[97]. C57BL/6 J WT and
TCRδ[-/-] mice were given buprenorphine HCl 0.1 mg/kg sub-
cutaneously once 1 h prior to surgery. The procedure was performed in
a biosafety hood while mice were placed on a heat pad and anesthesia
was provided throughout the procedure. Mice were fixed on a ste-
reotaxic frame and anesthetized with isoflurane (4%). Moderate TBI
was induced using the CCI Impact One device (Leica Biosystems).
Sterile surgical equipment was used. First, the dorsal part of the head
was shaved and sterilized followed by a midline skin incision. Skin was
retracted exposing the surface of the skull. Craniotomy was performed
by removing a 3-mm skull bone overlying the right parietal cortex. A
1.5 mm metal impactor tip was extended and lowered at a 20° angle at
the following coordinates: 1.8 mm caudal to bregma, 2.0 mm laterally.
The tip was retracted, and the electromagnetic device was triggered
using the following parameters: 1 mm depth, 5.0 m/s velocity and a
dwell time of 0.8 s. Following the impact, the bone flap was returned
and glued using a sterile bone wax. The skin was sutured using 6-0
black nylon nonabsorbable monofilament sutures as 3–5 interrupted
sterile sutures. Mice were allowed to recover on another heat pad and
then returned to new cages. Sham mice underwent the same proce-
dure including craniotomy but no brain injury impact. Instead, the
impact tip was discharged in the air.

## Generation of microglia single-cell suspension
Mice were intracardially perfused with ice cold Mg[2+] and Ca[2+] free 1X
Hank's balanced salt solution (HBSS; Gibco, #14175). Brains were gently
dissected from the overlying skull with the peri-contusion brain tissue
quickly removed and placed on ice. Brains were mechanically homo-
genized using Dounce homogenizers in ice cold HBSS. Brains were
homogenized 10 times each with the loose and tight pestles while
simultaneously rotating the pestle. The cell suspension was then
transferred to prechilled 15 mL tubes and passed through an HBSS pre-
wet 70 μm cell strainer (Fisher Scientific, #22363548). Cell suspensions
were then centrifuged at 300 g for 5 min at 4 °C. Debris and myelin
were removed using a modified room-temperature Percoll gradient[72].
Briefly, cell pellets were resuspended in 5 mL room-temperature 30%

Percoll Plus cell separation liquid media (GE Healthcare Biosciences,
#17-5445-01) diluted in HBSS and then spun for 20 min at 350 g with
acceleration rate of 5 and deceleration rate of 3. Microglia cells sub-
sequently pellet at the bottom of the 15 mL tubes with myelin and
debris floating above Percoll Plus solution. The cell pellet was washed
with 10 mL ice cold FACS buffer (Mg[2+] and Ca[2+] free HBSS with 2% FBS,
0.4% EDTA 0.5 M and 2.5% HEPES 1 M) and spun again for 5 min at
300 g at 4 °C. All samples were then resuspended in 100 μl of ice cold
FACS buffer containing APC/Cy7 anti-CD45 (Biolegend, #103116,
1:100), PE/Cy7 anti-CD11b (Biolegend, #101216, 1:100), FITC anti-Ly-6C
(Biolegend, #128006, 1: 200) antibodies as well as the microglia spe-
cific 1:800 APC anti-4D4 antibody[61,62] provided by Oleg Butovsky. Cells
were stained for 20 min on ice. Samples were then washed in 10 mL ice
cold FACS buffer and spun for 5 min at 300 g and then resuspended in
200 μl of ice cold FACS buffer. 1000 live CD45[low]CD11b[pos]Ly-
6C[neg]4D4[pos] microglial cells were sorted on a BD FACSAria II using
the 70 μm nozzle with a sort speed of approximately 10,000 events
per second. Dead cells were excluded using 7-AAD viability staining
solution (Biolegend, #420404, 1:100). Gating strategy for microglia
sorting is shown in Supplementary Fig. 9a, b. The cells were sorted into
prechilled 1.5 mL tubes containing 1% 2-mercaptoethanol in TCL buffer
(Qiagen, #1031576). After sorting, the cells were immediately short-
spun, placed on dry ice and stored in −80 °C freezer.

## RNA sequencing
At the end of FACS sorting, single cell suspensions were plated in a
prechilled 96-well twin.tec PCR plate LoBind, full skirted plate
(Eppendorf, #0030129512) and shipped to the Broad Institute for
Smart-seq2 RNA sequencing[98]. Samples were processed for cDNA
generation and Illumina Nextera XT library construction. Sequencing
data was generated using 2×38 bp paired end sequencing on the
NextSeq500. Transcript-level gene expression analysis of the raw RNA-
seq reads was performed as previously described[99]. First, quality of the
raw RNA-seq reads was assessed using FastQC quality control tool for
high throughput sequence data[100]. Reads were concatenated then
trimmed using Trimmomatic[101]. The RNA reads were aligned to the
mouse mm10 reference genome using HISAT2[102]. HISAT2-generated
SAM files were sorted and converted into BAM files using Samtools[103].
The sorted reads were assembled into transcripts using StringTie[104].
Next, StringTie-generated transcript lengths and abundance estimates
were converted into count matrices using Tximport[105]. Differential
gene expression analysis was performed with false discovery rate
(FDR)-adjusted $P$ values using DESeq2 with an adjusted $P$ cutoff value
of 0.05[106]. Data visualization was performed in R (version 4.0.3).
Heatmaps and clustering were generated using heatmap.2 from the
gplots package. For clustering, the z-scores were calculated using the
mean expression of biological replicates per disease stage/condition
and then subsequently clustered using K-means. Two-dimensional
principal component analysis (PCA) plots were generated using the
ggplot2 package and prcomp from the stats package. MA plots were

generated using ggmaplot from the ggpubr package. Pathway analysis was performed using gene set enrichment analysis (GSEA)[107,108]. GSEA or GSEA Preranked analyses were used to generate enrichment plots for RNA-seq data using MSigDB molecular signatures for canonical pathways: hallmark (h.all) and gene ontology (c5.cp.all) pathways. From the enrichplot package, enrichment plots were generated using gseaplot2, dot plots were generated using dotplot and gene networks were visualized using cnetplot. Statistical analysis in GSEA was determined by one-tailed $t$-test in GSEA. In all cases, FDR values were derived using the Benjamini-Hochberg test and $P$ values were calculated. To identify regulators of gene expression networks, ingenuity pathway analysis (IPA) software (Qiagen) was used by inputting gene expression datasets. 'Canonical pathways' and 'upstream analysis' metrics were considered significant at $P < 0.05$. To identify regulatory networks, once a specific regulator was identified, the 'Build>Grow' function was used to identify molecules of the selected network. Statistical analysis using Qiagen IPA was carried out with a right-tailed Fisher's exact test.

## Flow cytometry

Microglia were separated using Percoll Plus as described above and spleen, deep cervical lymph nodes (cLN) and blood were harvested before intracardiac perfusion. All tissues were taken from the same animal. cLN and spleens were macerated through a 70 µm cell strainer (Fisher Scientific, #22363548) and centrifuged at 300 g for 5 min at 4 °C. For spleens, RBCs were lysed using ACK lysing buffer (eBioscience, #A1049201). Both cLN and spleen cells were washed in ice cold FACS buffer prior to staining. Blood was initially collected in Vacutainer heparinized tubes (BD Biosciences, #366664) then stained and lysed using FACS Lysing Solution according to manufacturer's instructions (BD Biosciences, #349202, 1:10). For intracellular cytokine staining, cells were stimulated for 4 h with 50 ng/mL phorbol 12-myristate 13-acetate (PMA; Sigma-Aldrich, #P1585), 1 µM ionomycin (Sigma-Aldrich, #I3909) and 1 µg/mL GolgiStop protein-transport inhibitor containing monensin (BD Biosciences, #554724) diluted in Iscove's Modified Dulbecco's Medium (IMDM; Gibco, #31980030) containing heat-inactivated 10% fetal bovine serum (FBS; Gibco, #10438026), 100 U/mL penicillin-streptomycin mixture (Lonza, #DE17-602E), 55 µM 2-mercaptoethanol (Gibco, #21985023) and 1% non-essential amino acids (Lonza, #BE13-114E) prior to staining with antibodies. Fc receptors were blocked with anti-mouse CD16/CD32 (Bio X Cell, #BE0307, 1:50) for 15 min on ice. Surface markers were stained for 20 min at 4 °C in FACS buffer then fixed and permeabilized with Foxp3/Transcription Factor Staining Buffer Set (eBioscience, #00-5523-00). Cells were then stained for intracellular cytokines and washed in FACS buffer. Zombie Aqua Fixable Viability Kit (Biolegend, #423102, 1:1000) was used to exclude dead cells. Flow cytometric acquisition was performed on LSRFortessa and FACSSymphony A5 (BD Biosciences) using DIVA software (BD Biosciences) and data were analyzed with FlowJo software version 10 (TreeStar Inc.). The staining antibodies used are AF700 anti-CD45 (Biolegend, #103128, 1:300), BV785 anti-CD11b (BD Biosciences, #740861, 1:300), BV605 anti-CD3ε (Biolegend, #100351, 1:300), APC anti-TCRγδ (eBioscience, #17-5711-82, 1:300), FITC anti-Vγ1.1/Cr4 (Biolegend, clone 2.11, #141104, 1:300), PE anti-Vγ2 ("anti-Vγ4", Biolegend, clone UC3-10A6, #137706, 1:300), BV421 anti-IFN-γ (Biolegend, #505830, 1:300), PE/Cyanine7 anti-IL-17A (Biolegend, #506922, 1:300), BUV661 anti-CD45 (BD Biosciences, #565079, 1:1000), BB515 anti-CD11b (BD Biosciences, #564454, 1:1000), APC-Fire 750 anti-Ly-6C (Biolegend, #128046, 1:800), BUV563 anti-Ly-6G (BD Biosciences, #565707, 1:200), PE/Dazzle594 anti-CX3CR1 (Biolegend, #149013, 1:400), BB700 anti-CCR2 (BD Biosciences, #747965, 1:400–stained separately at 37 °C for 15 min), BV605 anti-I-A/I-E ("MHC class II", BD Biosciences, #563413, 1:800), BV650 anti-CD3ε (BD Biosciences, #564378, 1:200), BUV496 anti-CD4 (BD Biosciences, #612952, 1:400), BUV805 anti-CD8 (BD Biosciences, #564920, 1:400), FITC anti-FoxP3 (eBioscience, #11-5773-82, 1:200), PE anti-LAP (Biolegend, #141404,

1:100), PE/Dazzle 594 anti-IL10 (Biolegend, #505034, 1:100), Fixable Viability Dye eFluor 506 (eBioscience, #65-0866-18, 1:800).

## Brain edema

Brains were removed at 72 h after CCI and the ipsilateral hemisphere was weighed (wet weight). The hemisphere was then dried at 60 °C for 48 h, and dry weights were obtained. The percentage of brain water content was expressed as (wet-dry weight)/wet weight × 100, as previously described[109].

## Immunostaining

Mice were intracardially perfused with 10 mL ice cold 1× phosphate buffered saline (PBS) followed by 10 mL ice cold 4% paraformaldehyde (PFA). Brains were gently dissected, post-fixed in 4% PFA overnight at 4 °C and dehydrated in 30% sucrose for 3 days at 4 °C. Brains were then paraffin embedded and 5-µm sections were obtained by a microtome and stored at room temperature. All immunostaining was performed on the Leica Bond III automated staining platform using the Leica Biosystems Refine Detection Kit (Leica Biosystems, #DS9800). Sections were incubated with the following primary antibodies: rabbit monoclonal anti-APP (Abcam, #ab32136, 1:4000, EDTA antigen retrieval), mouse monoclonal anti-ptau Ser202/Thr205 AT8 (eBioscience, #MN1020, 1:100, citrate antigen retrieval), rabbit anti-Iba1 polyclonal (Wako Chemicals, #019-19741, 1:500, EDTA antigen retrieval), rabbit anti-iNOS polyclonal (Abcam, #ab15323, 1:100, EDTA antigen retrieval). For immunofluorescence, the following fluorophores were used: Alexa Fluor 488 Tyramide (Life Technologies, #B40953), Alexa Fluor 555 Tyramide (Life Technologies, #B40955), Alexa Fluor 594 Tyramide (Life Technologies, #B40957) and Alexa Fluor 647 Tyramide (Life Technologies, #B40958). Nuclei were counterstained with NucBlue Fixed cell stain (Life Technologies, #R37606). Sections were sealed and cover-slipped with mounting media (Invitrogen, #P36961). Hematoxylin and eosin (H&E) staining was done using mercury-free Harris Hematoxylin (Fisherbrand, #23-245-678) and alcoholic Eosin-Y (Anatech, #832). Images were acquired using a Leica DMi8 widefield microscope (Leica Biosystems) using a 20× objective with Leica LAS AF software and processed using Fiji and LAS X. For Sholl's analysis, we used the Sholl's analysis plug-in in Fiji to determine the number of microglial intersections measured between 10 µm and 60 µm from the soma.

## Quantitative PCR

Freshly dissected brain tissue was stored in RNAlater (Sigma-Aldrich, #R0901) at 4 °C and processed the following day. Brains were then electrically homogenized (Kinematica, #PT1200E) in buffer RLT (Qiagen, #79216) containing 1% 2-mercaptoethanol. Lysate was centrifuged for 3 min at maximum speed at 4 °C. Lysate was transferred to a genomic DNA eliminator column and RNA extraction was performed as per manufacturer's instructions (RNeasy Plus Mini Kit, Qiagen, #74134). For RNA extraction of freshly FACS-sorted cells, RNeasy Plus Micro Kit, (Qiagen, #74034) was used. Next, RNA was reverse-transcribed using the High-Capacity cDNA Reverse Transcription Kit with RNase Inhibitor (Life Technologies, #4374966). Quantitative real-time PCR was then performed on the cDNA using TaqMan Fast Universal PCR Master Mix (2X) no AmpErase UNG (Life Technologies, #4352046) with a Vii 7 real-time PCR system (Applied Biosystems) with the following primers and probes: Il1b (Mm00434228_m1), Il6 (Mm00446190_m1), Il10 (Mm01288386_m1), Il12a (Mm00434169_m1), Il13 (Mm00434204_m1), Il17a (Mm00439618_m1), Il18 (Mm00434226_m1), Il23a (Mm00518984_m1), Il33 (Mm00505403_m1), Ifng (Mm01168134_m1), Tnf (Mm00443258_m1), Tgfb1 (Mm01178820_m1), Foxp3 (Mm00475162_m1), Ccl2 (Mm00441242_m1), Ccl5 (Mm01302427_m1), Ccl20 (Mm01268754_m1), Cxcl10 (Mm00445235_m1), Gata3 (Mm00484683_m1), Ccr4 (Mm01963217_u1), Ccr5 (Mm01963251_s1), Ccr6 (Mm99999114_s1) and Ifngr1 (Mm00599890_m1) in addition to in-

house-designed (and synthesized by Life Technologies) probes targeting Vγ1 (AI89KL3) and Vγ4 (AIBJXX4) chain segments. Quantitative PCR data were analyzed by the delta-delta Ct method by normalizing the expression of each gene to *Gapdh* (Mm99999915_g1).

## Western blotting

Freshly dissected brain tissue was electrically homogenized (Kinematica, #PT1200E) in RIPA lysis and extraction buffer (eBioscience, #89900) containing 1% Halt protease and phosphatase inhibitor cocktail (eBioscience, #78441). Samples were kept on ice for 5 min and protein lysates were then obtained by centrifugation at maximum speed for 10 min at 4 °C. Protein quantification was performed by Nanodrop (eBioscience). Equal amount of total protein was obtained by dilution into 4× Bolt LDS Sample Buffer (Invitrogen, #B0007), 10× Bolt Sample Reducing Agent (Invitrogen, #B0009) and deionized water up to a total volume of 40 μl. Lysates were boiled at 70 °C for 10 min. Next, samples along with a SeeBlue Plus2 pre-stained protein standard (Invitrogen, #LC5925) were loaded into Bolt 4−12% Bis-Tris mini protein gels (Invitrogen, #NW04125BOX) and SDS-PAGE was performed at 120 V in diluted 20× Bolt MES SDS running buffer (Invitrogen, #B0002). Proteins were blotted to the transfer membrane using the iBlot dry blotting system (Invitrogen, #IB1001) and nitrocellulose mini iBlot Transfer Stack (Invitrogen, #IB301002). Membranes were then cut horizontally according to the molecular weights of the target proteins and guided by the protein standard ladder. Membranes were blocked for 30 min on a shaker using StartingBlock T20 tris-buffered saline (TBS) blocking buffer (eBioscience, #37543) that already contains 0.05% Tween-20. Membranes were then washed 3 times with the blocking buffer. Primary antibodies were diluted in blocking buffer and incubated with the membranes overnight at 4 °C. Primary antibodies used were: rabbit monoclonal anti-HMGB1 (Cell signaling, #6893 S, 1:1000), rabbit polyclonal anti-ZO1 (Abcam, #ab96587, 1:1000), rabbit monoclonal anti-Occludin (Abcam, #ab167161, 1:2000) and rabbit monoclonal anti-GAPDH (Cell signaling, #5174 T, 1:2000). Membranes were then washed 3 times and incubated with the secondary antibody horseradish peroxidase (HRP)-linked goat anti-rabbit (Cell signaling, #7074P2, 1:2000) on a shaker for 1 h. The membranes were washed again, incubated with Pierce ECL Plus substrate (eBioscience, # 32134) and exposed on iBright CL1500 imaging system (Thermo Fisher Scientific). Images were imported into Fiji and band intensity was quantified relative to GAPDH.

## Depletion of γδ T cell subsets

C57BL/6 J WT mice were depleted of their Vγ1 cell, Vγ4 cells as well as total TCRγδ cells using 200 μg anti-Vγ1 (Bio X Cell, clone 2.11, #BE0257), 200 μg anti-Vγ2 ("anti-Vγ4", Bio X Cell, clone UC3-10A6, #BE0168) and 200 μg anti-TCRγδ (Bio X Cell, clone UC7-13D5, #BE0070) monoclonal antibodies, respectively. The monoclonal antibodies were injected intraperitoneally 24 h before TBI.

## Purification and adoptive transfer of γδ T cell subsets

Splenocytes were isolated from C57BL/6 J WT mice as described above. γδ T cells were first purified and enriched using TCRγδ T cell isolation microbeads kit (Miltenyi Biotech, #130-092-125) on a magnetic MACS separator prior to sorting. Eluted cells were pooled and stained in FACS buffer containing APC/Cy7 anti-CD45 (Biolegend, #103116, 1:100), PE/Cy7 anti-CD3ε (Biolegend, #152313, 1:100), APC anti-TCRγδ (eBioscience, #17-5711-82, 1:300), FITC anti-Vγ1.1/Cr4 (Biolegend, clone 2.11, #141104, 1:300) and PE anti-Vγ2 ("anti-Vγ4", Biolegend, clone UC3-10A6, #137706, 1:300) for 20 min on ice. Dead cells were excluded using 7-AAD viability staining solution (Biolegend, #420404, 1:100). CD45$^{pos}$CD11b$^{neg}$CD3ε$^{pos}$TCRγδ$^{pos}$Vγ1$^{pos}$Vγ4$^{neg}$ and CD45$^{pos}$CD11b$^{neg}$CD3ε$^{pos}$TCRγδ$^{pos}$Vγ1$^{neg}$Vγ4$^{pos}$ subsets were sorted using FACSAria II (BD Bioscience). Gating strategy for cell sorting is as shown in Supplementary Fig. 9c. TCRγδ cell subsets were sorted and immediately transferred into TCRδ$^{-/-}$ mice. Each mouse received 5 × 10$^5$ cells intravenously at the same time of TBI.

## Transwell culture

Microglia and γδ T cell subsets were isolated as described above. Microglia were directly sorted into a microglia culture media composed of 10% fetal bovine serum (FBS; Gibco, #10438026), 100 U/mL penicillin-streptomycin mixture (Lonza, #DE17-602E), 55 μM 2-mercaptoethanol (Gibco, #21985023), 1% non-essential amino acids (Lonza, #BE13-114E) supplemented in Dulbecco's Modified Eagle Medium (DMEM)/F-12 Glutamax media (Gibco, #10565018). γδ T cells were directly sorted into a lymphocyte culture media composed of 10% fetal bovine serum (FBS; Gibco, #10438026), 100 U/mL penicillin-streptomycin mixture (Lonza, #DE17-602E), 55 μM 2-mercaptoethanol (Gibco, #21985023), 1% sodium pyruvate (Lonza, #BE13-115E) and 1% HEPES (Lonza, #BE17-737E) supplemented in Roswell Park Memorial Institute (RPMI) 1640 media (Gibco, #11875119). Cell count was performed and 100,000 microglia cells per condition were plated in the lower chambers of 24-well plates (Corning, #3524) and 50,000 γδ T cells per condition were plated in 0.4 μm inserts (Millipore, #MCHT24H48). 1 μg/mL of anti-CD3 (Biolegend, #100340) was added to all inserts. According to the experimental condition, 10 μg/mL of anti-TGF-β (Bio X Cell, clone 1D11.16.8, #BE0057) or anti-IL-17a (Bio X Cell, clone 17F3, #BE0173) were added to the inserts. Cells were co-cultured at 37 °C in a humidified incubator with 5% CO$_2$ for 48 h after which microglia was taken for RNAseq.

## Lentivirus generation and injection

Third-generation lentivirus packaging was provided by AMSBIO (Massachusetts, USA). Briefly, a transfer plasmid carrying the gene of interest "Ifngr1" was co-transfected with a proprietary envelope plasmid encoding VSV-G and packaging plasmids encoding Gag/Pol and Rev into HEK293T packaging cells. CD11b was used as a promoter and emGFP was inserted to act as a fluorescent protein. After 48 h of incubation, the supernatant is collected and centrifuged to remove cell debris and then filtered. Lentiviral particles were subsequently concentrated with polyethylene glycol. Lentivirus titer was measured using the p24 ELISA method. Briefly, HIV-1 p24 antigen was captured by anti-p24 coated microtiter wells and sandwiched with biotinylated secondary anti-p24 antibody. Subsequently, a streptavidin-HRP conjugate and a substrate were added. Color intensity was measured spectrophotometrically to indicate the level of p24 in the sample, which was then quantified against a p24 standard curve. p24 value was then correlated to virus titer. The pellet containing lentivirus was resuspended in PBS and aliquots were stored at −80 °C. Quality control steps included titer measurement, sterility testing for bacteria and fungi, mycoplasma detection, fluorescent protein transduction test and drug selection test. The vector without insertion of mi-shIFN-γR1 was used as control.

Mice were injected with approximately 1 × 10$^7$ IU in the bilateral ventricles using a 25-μl Hamilton syringe (Sigma-Aldrich, #20787) on a stereotaxic alignment system at the following coordinates: 1.0 mm caudal to bregma, 2.0 mm laterally and 2.5 mm below the skull surface. Injection speed was maintained at 1 μl/min to prevent leaking.

## Behavioral testing

All behavioral experiments were carried out during daylight hours in a blinded fashion. Mice were allowed to acclimate for 3−4 weeks prior to testing. At the day of testing, animals were acclimatized to the behavioral rooms for at least 30 min. Animals from different groups were tested consecutively. For the depletion and adoptive transfer experiments, mice underwent all behavioral assays beginning with the rotarod test, followed by the open field test and then the Morris water maze test. To account for social interactions, sham and TBI mice of the same genetic background were mixed in the same cage. With the

exception of rotarod and marble burying tests, data were digitally recorded using a camera-enabled Noldus EthoVision XT software v.17.0.

## Rotarod test

Motor coordination was assessed using a rotarod (Harvard Apparatus, Massachusetts, USA). Rotarod was set to accelerate from a speed of 5–60 rotations per min (rpm) in a 180-s time trial. Each mouse was given an exposure trial (4 rpm, 60 s) to familiarize the animal with the task. The exposure trial was not included for data analysis. Latency to fall from the rod (or cling to and rotate with the rod for three consecutive rotations) was recorded for three trials on each testing day, allowing a 5-min of rest with access to food and water between each trial. A mouse that remained on the rod for the entire 3 min was assigned a score of 180 s. Scores from the three trials on each testing day were averaged to give a single score for each mouse.

## Open field test

Anxiety and exploratory behavior were assessed using the open field test. Four mice, selected from different experimental groups, were tested at once for a duration of 10 to 15 min. The open field was a 40 × 40 cm apparatus consisting of four 20 × 20 cm arenas with opaque walls. Mice were individually placed in the center of the arena and a camera located above the apparatus recorded activity. The software provided automated calculations of the percentage of time spent in the center of each arena, average velocity, total distance traveled and immobility (with mobility defined as at least 35% of the animal moving for at least 2 s).

## Marble burying test

Compulsive behavior was assessed using the marble burying test. The test was performed using new animal cages with additional autoclaved woodchip bedding filled to a depth of 2 cm. Twenty black glass marbles were placed on the surface of the cage bedding in a rectangular shape of four rows each with five marbles equidistant from one another. After a 15-minute testing session, mice were gently removed from the testing cages, and the number of marbles buried to at least a two thirds depth was counted by a blinded investigator.

## Morris water maze

Spatial learning and memory were assessed using the Morris water maze test[110]. The test consisted of 1 visible trial, and 4 invisible trials followed by 1 probe trial. Trials were conducted on consecutive days. The pool was a circular white tank 122 cm in diameter, filled with water (23 ± 1 °C) to a depth of approximately 30 cm. Each mouse was allowed three attempts with the exception of the probe trial during which each mouse was tested only once. For the visible trial on day 1, a transparent acrylic platform 10 cm in diameter was placed 1 cm above the water surface and was located approximately 15 cm from the edge of the maze. The platform location was changed with every visible trial. For the invisible trials on days 2, 3, 4 and 5 white paint was added, and the platform was submerged 1 cm below the surface of the water and kept in the same location throughout the four days. For the probe trial on day 6, the platform was completely removed. Visual clues of different colors and shapes were posted on the room walls. The water maze was divided into four equal quadrants and a ceiling camera directly over the water maze was recording animal activity. Four start positions were designated along the perimeter of the tank, and the order in which these positions were used was changed each day. For each trial, the mouse was gently placed in the designated start position facing the wall. Mice were allowed 90 s to locate the platform using the spatial cues in the room, after which the animal remained on the platform for 15 s. Only during testing days 1, 2 and 3, If the mouse did not find the platform in the allotted 90 s, it was gently guided to the platform and allowed to remain there for 15 s and received the maximum score of 90 s for that trial. After each trial, mice were towel dried and placed into a heated cage. Latency to platform, percentage of time spent in target quadrant, frequency of target quadrant transitions and swimming velocity were recorded.

## Microbial community analysis

Fecal pellets were collected from WT and TCRδ$^{-/-}$ male mice. DNA was extracted using the Qiagen DNAeasy Powerlyzer Kit (Qiagen, #12855). A microbiome sequencing library was constructed by amplifying the V45 region of the rRNA 16 S gene, according to methods adapted from the Earth Microbiome Project and as we have previously reported[55]. QIIME2[111] was used to analyze rRNA 16 S seq, including denoising and dereplicating with DADA2, taxonomy was assigned using the EZTaxon EZ-biocloud, release May 2018 formatted for QIIME[112] trained on the V45 region using the RDP classifier. Samples were removed if they had fewer than 1000 reads and ASVs were removed if they had fewer than 10 reads or were in fewer than 2 samples. β-diversity was assessed by PCoA plots and PERMANOVA based on weighted and unweighted UniFrac distances. Changes in relative abundance was determined by linear discriminant analysis effect size (LEfSe)[113].

## Statistics & reproducibility

Statistical analysis was performed using GraphPad Prism 9 software. Data distribution was assumed to be normal, but this was not formally tested. Data are presented as mean ± sem and two-tailed Student's $t$ tests (unpaired) or ANOVA multiple comparison tests were used to assess statistical significance. Bonferroni-adjusted $t$-test values were used to determine significant differences between groups. Data for each experiment were collected and processed randomly and animals were assigned to various experimental groups randomly as well. No statistical methods were used to predetermine sample size. Sample sizes were chosen in accordance with previous studies in the field[114,115]. The investigators were blinded to allocation during experiments and outcome assessment as mentioned in the Behavioral testing section. Each experiment was repeated 2–3 times. All $n$ and $P$ values and statistical tests are indicated in figure legends.

## Reporting summary

Further information on research design is available in the Nature Portfolio Reporting Summary linked to this article.

## Data availability

RNA-seq data samples were deposited into the Sequence Read Archive (SRA) of the National Center for Biotechnology Information (NCBI), BioProject accession number PRJNA936187. Source data are provided as a Source Data file. Source data are provided with this paper.

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

## Acknowledgements

This work was funded by NIH grant number R21AG063187-01A1 awarded to Dr. Howard Weiner, and in part, by a Brigham Research Institute microgrant awarded to Dr. Hadi Abou-El-Hassan. We thank the Neuro-Technology Studio at the Brigham and Women's Hospital for providing instrument access and consultation on data acquisition and data analysis. We thank Dana-Farber/Harvard Cancer Center in Boston, MA, for the use of the Specialized Histopathology Core, which provided histology and immunohistochemistry service. Dana-Farber/Harvard Cancer Center is supported in part by an NCI Cancer Center Support Grant # NIH 5 P30 CA06516.

## Author contributions

H.A. designed and performed experiments and wrote the manuscript. R.M.R. designed and supervised experiments and wrote the manuscript. G.G., B.K.T., J.R.L., G.L.V.D.O., K.J.H., A.H.M., T.Y., and Z.Y. performed experiments. R.K. sorted cells for RNA sequencing experiments. R.M.R., L.M.C., S.I., and O.B. supervised the project and H.L.W. supervised the project and wrote the manuscript.

## Competing interests

The authors declare no competing interests.
