## [Peer Review File · Nature Communications]

V γ 1 and V γ 4 gamma-delta T cells play opposing roles in the immunopathology of traumatic brain injury in male miceREVIEWER COMMENTS

Reviewer #1 (Remarks to the Author):

The manuscript by Abou-El-Hassan et al has revealed the critical role of $\gamma\delta$ T cells in the pathogenesis of acute and chronic TBI. Mechanistically, the V γ 4 subset promotes inflammation by activating microglial cells via secreting IL-17 and IFN γ , while the V γ 1 subset attenuates inflammation by maintaining a homeostatic microglial signature in a TGF β -dependent manner. This study has entailed a set of experimental approaches, using the genetic $\gamma\delta$ knockout mice, antibody depletion, adoptive transfer and detailed transcriptomic analysis. Overall, this is an interesting study with a well-designed experimental strategy and largely supportive results. However, the following points need to be addressed and clarified to strengthen the findings.

1) According to PMID: 27656033, the percentage of microglia/macrophages expressing MHCII appears very low (\sim 0.2-1.5%) (Figs. 2f, 4f, 5f,6f, 7f). This needs to be clarified and confirmed.

2) Figs. 2h, 5l: It is not clear what are the comparisons for the indicated p values.

3) Fig. 4: It is not clear why depletion of total $\gamma\delta$ T cells did not significantly reduce CD8 and monocytes, improve behavioral parameters and induce changes of most of microglial genes compared to PBS controls, as those observed in $\gamma\delta$ T KO mice (Figs. 1-3)?

4) Fig 5: Transfer of V γ 4 cells did not change the behavioral parameters (Figs. 5i-n). This is not strong enough to support that V γ 4 $\gamma\delta$ T cells are detrimental. Fig. 5p: Fcrls and Itgb5 are not significantly increased in V γ 1 transfers.

5) Extended Fig. 8a: There are more V γ 4+IL17+ cells in the cLN of TBI, but fewer V γ 4+ cells in TBI compared to sham in the cLN (Extended Fig. 4B). The absolute numbers of V γ 4+IL17+ cells in the cLN will be low, which would argue against any significant changes induced by V γ 4+IL17+ cells. Please clarify this.

6) This manuscript focused on the crosstalk between $\gamma\delta$ T and microglia. $\gamma\delta$ T cells definitely influence CD4+, CD8+ and monocytes. To what extent the changes of these cells contribute to the overall effects of $\gamma\delta$ T knockout or ablation on acute and chronic TBI?

Reviewer #2 (Remarks to the Author):

Review: V γ 1 and V γ 4 $\gamma\delta$ T cell subsets play opposing roles in traumatic brain injury

This experimental study by Hadi Abou-El-Hassan et al. is well designed and of high interest for the readership of Nature Communication. It addresses fundamental and so far unknown immune mechanisms in the case of traumatic brain injury (TBI) reflecting still a high scientific and clinical challenge.

As already stated in the introduction part of the manuscript, there is almost no knowledge about the role of $\gamma\delta$ T cells in the context of traumatic TBI, resulting in a current lack of therapeutic strategies.

A hypothesis was not defined. The aim of the study was to investigate the $\gamma\delta$ T cells and provide corresponding mechanisms involved in the acute (days) and chronic (year) immune response post TBI. Whether these findings lead to novel therapeutic approaches remains speculative.

Summary of the key results:

The study demonstrates that $\gamma\delta$ T cells drive and modulate the pathophysiology of TBI and the neurological functions as early as one day and up to one year.

The clinically relevant mouse model of a controlled cortical impact revealed in TCR δ -/- mice signs of reduced tissue damage and reduced inflammatory response accompanied by an improvement of a variety of neurocognitive functions.

Furthermore, a balanced opposing response is suggested for V γ 1 and V γ 4 $\gamma\delta$ T cell subsets with

their corresponding mediator pattern during TBI progression.

An adoptive transfer of V γ 1 and V γ 4 $\gamma\delta$ T cell subsets from either WT mice or TCR δ $-/-$ mice at the time of injury revealed differential effects.

Furthermore, the tight interaction of $\gamma\delta$ T cell subsets with the microglia is demonstrated on a functional and transcriptomic level, and suggest, that V γ 1 $\gamma\delta$ T cells reduce microglia activation and TBI-associated neuroinflammation.

Materials and Methods and data presentation:

An established TBI mouse model was used and includes sufficient references for the chosen parameters. It contains all essential control animals and appropriate biological and technical replicates resulting in reproducible and reliable data.

The methods used in the study are accurate, diverse and well- described in the method section.

Appropriate statistical methods were used to evaluate the data.

Data-presenting graphs are well-designed and captured, figure legends include further information for clear understanding.

All in all, the manuscript presents a well-structured study of high scientific value. There are only some minor concerns:

1. The authors state in the abstract and introduction part that the role of $\gamma\delta$ T cells is "unknown" or "has not been explored" in the context of TBI. However, although little is known there is for example one study also on a murine TBI model, which addressed $\gamma\delta$ T cells at least in the spleen after TBI (e.g. Richard P Tobin et al. : Traumatic brain injury causes selective, CD74-dependent peripheral lymphocyte activation that exacerbates neurodegeneration. *Acta Neuropathologica Communications* 2014, 2:143). The authors should reword their statements a little more careful.

2. The authors also consider the $\gamma\delta$ Tcells the "first line of defense" (page 4, line 72). Commonly, neutrophils and monocytes and the complement system are defined the "first line of defense" so the authors should rather introduce $\gamma\delta$ T cells as contributing "part of the first line of defense".

3. Have the authors defined histological changes beyond only determination of the "lesion volume" (which was significantly reduced in the TCR δ $-/-$ mice)? It would be helpful for clinical translation to patho-morphologically assess at an early time point e.g. the degree of hematoma, neutrophil influx, fibrin deposition, monocyte/macrophages infiltration, thrombi formation, and especially signs of edema? In principal, these morphological findings could also be different in the TCR δ $-/-$ mice early after TBI ?

4. A graphical abstract -if agreed upon by the editor- could be very helpful for the reader to get these comprehensive data with many transcriptomic and adoptive transfer data condensed to the most important insights.

5. Mislabelling: page 12 line 264: Fig. 6 i-n should read Fig. 5 i-n.

6. Could the authors please provide in the discussion part some information on the limitations of the study or are there none?

Reviewer #3 (Remarks to the Author):

In this manuscript, Habou-El-Hassan et al report an opposite role for V γ 1 and V γ 4 gd T cells in a model of Traumatic Brain Injury (TBI). Using neutralizing monoclonal antibodies and adoptive cell transfer, the authors demonstrate that the V γ 4 subset promotes TBI pathology through the production of IL-17 and IFN-g, while the V γ 1 subset appears protective, maintaining the microglia homeostasis through the production of TGF-b.

Although mostly expected given the emerging roles of gd T cells in neuroinflammation, these findings are important and point at potential target for the development of immunotherapeutic strategies for the treatment of TBI and related pathologies. There are however a number of major shortcomings regarding the strength of some data and associated mechanisms, as well as the

organization of the manuscript in general. The following issues should be addressed in order to ensure that the study is suitable for publication.

1- One of the main caveat of the study is the fact that the authors did not use littermates controls, especially for behavioural analysis. I could not find mention of littermates or at least, co-housing equilibration (>4 weeks) for the KO animals. When possible, mice should be derived from a Het X Het cross and WT littermates should be the appropriate controls. The importance of using WT littermates with as controls is crucial because there could be subtle differences in mouse housing conditions, handling and background genetics that may impact on overall immune profile at steady state and upon inflammation.

2- The mice used are all males. Would the authors expect to reproduce their findings with female mice? Again, there may be subtle sex biased differences in animal behavior that should be at least discussed. Also, how were the group set? Are all the mice going through all the behavior assays? If so, in which order? This should be specified as this could also induce confounding variables, the mice potentially becoming less anxious along the sequence of the tasks. While some tests completely make sense in the context of TBI and current literature, the rationale for using some others and their resulting data are obscure to me. For example, why running the marble burying test? Can the authors comment on the higher compulsive behavior observed in TCRdKO one year post TBI? Some differences can be observed between Sham WT and TCRdKO, consistently with the previously described pro-anxiety role of gd T cells (de Lima et al, 2020). The authors should acknowledge these basal differences and discuss how they would (or not) be conserved upon TBI.

3- The authors observe a sizeable population of Vg1-Vg4- T cells, without commenting on it. Do the frequency of this subset vary upon TBI? Are these cells Vg6+? Or even Vg7+, referring to Benakis et al, 2016? Then, would they be influenced by the microbiota? In this same line, what is the population of CD11b+Ly6Cint cells that clearly appears in every dot plots, from Figure 2 to Figure 5, and seems to vary according to the conditions and models used? Can they authors quantify and comment on these? More generally, the cell frequencies obtained by FACS analysis would be greatly strengthened by displaying associated absolute cell counts.

4- Previous studies report a dual role for IL-17 in a model of stroke, triggering the onset of neuroinflammation (Benakis et al, 2016) while promoting neural recovery after 14 days (Lin et al, 2016). In such context, it would be interesting to discuss potential windows on which these different gd T cell subsets could intervene and regulate TBI time-course and severity. As proof of concept, different timepoint of in vivo cell transfers or manipulation of the microglia cultures could have been tested.

5- Regarding the in vivo cell transfer experiments, it would be interesting to isolate the Vg1 and Vg4 subsets from TBI mice, besides naïve animals. Are they expected to display a different outcome? In other words, how would their functional properties being shaped by the TBI environment? Can the authors speculate on where and how would they be primed? In this line, don't Vg1 also produce IL-17 and IFN-g upon TBI? Why not? What could make switch their typical type 1 effector program towards the secretion of TGF-b?

6- Can they authors comment on how would Vg1 or Vg4 influence BBB permeability? Could they use an alternative and functional way to characterize it? How Vg1 infiltrate the brain if the BBB is intact?

7- Figure 3 is really difficult to understand. It needs to be reorganized, as it is hard to follow the panel numbering. For a clarification purpose, some data could be transfer to supplementary information. In general, if Vg4 and Vg1 have opposing role, it is not very intuitive to analyze the microglia phenotype in the presence or absence of total gd T cells, using a full KO strategy. I understand the general unbiased screening approach but this should rather be presented in supplementary data, with some data being then exploited/validated in mice treated with anti-Vg4 and anti-Vg1, as done in Fig 4o. On the hand, Extended Data Fig. 3g and h would be important to appear in the main figure.

8- Overall, it is difficult to follow the track of the genes that are highlighting from one figure to another, referring to the neuroinflammatory / homeostatic microglial status. Also, more explanations should be given to establish the link between this microglial status post-TBI and the other neurodegenerative conditions (line 165 to 172). I do not understand how to define "a chronic TBI-associated microglia signature" by linking the TBI signature to the one observed in other neurodegenerative diseases. Then, what about the association between the "acute TBI microglial transcriptome" and AD, PD, ALS (Fig 3n)?

Minor comments and proofreading:

- Line 158: "Acute microglia modulation was predicted to be driven by both IFN- γ ($z = -1.765$) and IL-17A ($z = -1.965$) as upstream regulators (Fig. 3j)". This does not appear clearly.
- Line 188: "We found that microglia from aged TCR $\delta^{-/-}$ mice had only 84 DEGs vs. 1,879 DEGs in microglia from aged WT-TBI mice (Extended Data Fig. 3b-e)" Unclear sentence that would benefit some rephrasing. Can they authors develop on the potential implications?
- Line 200: "To investigate whether V γ 1 and V γ 4 $\gamma\delta$ T cell subsets play opposing roles in acute and chronic TBI, we first performed a kinetic study" . This would not define their roles. Please rephrase.
- Line 394: "gd T cells expand upon TBI in lymphatics": Ref 86 does not match this statement, please cite the original paper
- Lines 436 and 446: please refer to C57BL/6j not just C57BL/6 as it is the case in the rest of the material and method section.
- Line 647: typo "three 3" erase one of them.
- Line 672: "extended data fig4b" does not show the gating strategy for the V γ subset sorts.
- Line 852: the authors mention "BDP, breakdown products" that do not appear on the figure itself.
- Extended Data Fig1:
 - typo TCT $^{-/-}$ that needs to be replaced by TCR.
 - Scales on the microcopy picture are missing.
 - Lines 859-861: sham are not shown, remove them from the legend
 - Lines 870-872: sham are not shown, remove them from the legend
 - Line 1021: sham are not shown, remove them from the legend
- Fig4g and Fig5g: scales are missing

V γ 1 and V γ 4 $\gamma\delta$ T cell subsets play opposing roles in traumatic brain injury

We thank the reviewers for the thorough critique of our study. Applied changes are shown in red font in the revised manuscript.

Response to Reviewers

Reviewer #1:

The manuscript by Abou-El-Hassan et al has revealed the critical role of $\gamma\delta$ T cells in the pathogenesis of acute and chronic TBI. Mechanistically, the V γ 4 subset promotes inflammation by activating microglial cells via secreting IL-17 and IFN γ , while the V γ 1 subset attenuates inflammation by maintaining a homeostatic microglial signature in a TGF β -dependent manner. This study has entailed a set of experimental approaches, using the genetic $\gamma\delta$ knockout mice, antibody depletion, adoptive transfer and detailed transcriptomic analysis. Overall, this is an interesting study with a well-designed experimental strategy and largely supportive results. However, the following points need to be addressed and clarified to strengthen the findings.

1) According to PMID: 27656033, the percentage of microglia/macrophages expressing MHCII appears very low (~0.2-1.5%) (Figs. 2f, 4f, 5f,6f, 7f). This needs to be clarified and confirmed.

Response: We thank the reviewer for stating this important observation pertaining to the percentage of microglia/macrophages expressing MHCII. In review of the publication by Ertürk *et al.* in The Journal of Neuroscience (PMID 27656033), the percentage of MHCII+ microglia is reported to be as high as around 8% in chronic ipsilateral TBI using clone NIMR-4 of an anti-MHCII staining antibody that is conjugated to the “PE” fluorophore. Reasons that would explain the relatively low percentage of MHCII expressing myeloid cells in our experiments compared to the paper by Ertürk *et al.* include differences in antibody clones and fluorophores used to stain MHCII and, most importantly, the lack of detailed gate strategy for MHCII-expressing microglia/macrophages in Ertürk *et al.* study, which does not allow us to accurately compare it between studies.

2) Figs. 2h, 5l: It is not clear what are the comparisons for the indicated p values.

Response: We thank the reviewer for this important comment. We have now clarified the *P* values in the corresponding legends of **Figures 2 and 5**, “with *P* values corresponding to TCR $\delta^{-/-}$ TBI versus WT-TBI at the corresponding distance from soma” was added to the legend of **Figure 2**; “with *P* values corresponding to WT-V γ 1 versus PBS” was added to the legend of **Figure 5**.

3) Fig. 4: It is not clear why depletion of total $\gamma\delta$ T cells did not significantly reduce CD8 and monocytes, improve behavioral parameters and induce changes of most of microglial genes compared to PBS controls, as those observed in $\gamma\delta$ T KO mice (Figs. 1-3)?

Response: We thank the reviewer for the comment. There are multiple reasons that may explain why treatment with anti-TCR $\gamma\delta$ did not yield a similar neuroinflammatory profile compared to that of the TCR $\delta^{-/-}$ TBI cohort. Briefly and as elaborated in the study limitations below (Reviewer 2,

point #6), the selection of antibody treatment dose (200 µg) was derived from what has been reported in the literature (PMIDs: 29426701, 23685781) rather than an in-house gradient-guided optimal concentration selection. While we do show successful reduction in the level of expression of Vγ1 and Vγ4 chains with anti-Vγ1 and anti-Vγ4, respectively (**Extended Data Fig. 5a**), it appears that anti-TCRγδ resulted in Vγ1 depletion more so than Vγ4 which may partially explain the findings reported in **Figure 4** in the anti-TCRγδ-treated cohort. Second reason would be, as importantly pointed out in point #6 below, that the neuroinflammatory milieu in TCRγδ^{-/-}TBI might be affected by CD4⁺, CD8⁺ as well as monocytes in a genetically modified mouse model. A third reason would be the fact that the anti-TCRγδ antibody has been shown to not necessarily deplete γδ T cells, but rather, internalize the TCR (PMID: 19130484). Thus, in this context, γδ T cells are still there, but may be functionally inactive or partially inactive. We are aware of those study limitations and have reported them in the manuscript as mentioned in our response to point #6 of Reviewer 2.

4) Fig 5: Transfer of Vγ4 cells did not change the behavioral parameters (Figs. 5i-n). This is not strong enough to support that Vγ4 γδ T cells are detrimental. Fig. 5p: Fcrls and Itgb5 are not significantly increased in Vγ1 transfers.

Response: We thank the reviewer for the comment. We have shown that treatment with anti-Vγ4 leads to a reduced neuroinflammatory profile in the acute phase (**Fig 4b-h**) as well as favorable neurocognitive outcomes in the chronic phase (**Fig 4i-n**). While adoptive transfer of Vγ4 γδ T cells worsened neuroinflammation (**Fig. 5b-g**), their phenotypic effect was unremarkable. There are few reasons that might explain why Vγ4 γδ T cells did not worsen behavioral function. First, studying the behavior at 1 month after TBI may be early to observe the phenotypic effect of Vγ4 cells. Second, it may be possible that the number of cells transferred is sufficient to induce a neuroinflammatory milieu but not enough to induce changes at the behavioral level. We have mentioned in the manuscript, lines 263-264, “In contrast, there was no change in neurocognitive deficits in Vγ4 recipients (**Fig. 5i-n**)”. We have mentioned the limited effect of Vγ4 cells on behavioral parameters among the study limitations (response to point #6 of Reviewer 2). Regarding *Fcrls* and *Itgb5*, we checked our analysis, and we apologize for the mistake, we have deleted them from the text.

5) Extended Fig. 8a: There are more Vγ4+IL17+ cells in the cLN of TBI, but fewer Vγ4+ cells in TBI compared to sham in the cLN (Extended Fig. 4B). The absolute numbers of Vγ4+IL17+ cells in the cLN will be low, which would argue against any significant changes induced by Vγ4+IL17+ cells. Please clarify this.

Response: We thank the reviewer for the comment. We hypothesize that TBI mobilizes cLN Vγ4 cells that migrate to the brain where they exert pro-inflammatory effects. Indeed, the lower percentage of Vγ4 cells in the cLN of TBI compared to Sham mice (**Extended Data Fig. 8a**) along with a higher percentage of Vγ4 cells in the brain (**Extended Data Fig. 4b**) may be consistent with our hypothesis. To clarify the cellular dynamics, the following was added to line 399 “concurrent with a decrease of Vγ4 cells in the deep cervical lymph nodes”. Despite the lower absolute number of IL-17+ cells, IL-17 has been shown to be a potent cytokine given its ability to induce the expression of chemokines (PMID 9531313) as well as its synergistic effect with other cytokines (PMID 14600152). Additionally, the residual Vγ4 cells remaining in the cLN reactively produced more IL-17 after TBI (**Extended Data Fig. 8a**).

6) This manuscript focused on the crosstalk between γδT and microglia. γδT cells definitely influence CD4+, CD8+ and monocytes. To what extent the changes of these cells contribute to the overall effects of γδT knockout or ablation on acute and chronic TBI?

Response: We thank the reviewer for the comment. Although the CD4⁺, CD8⁺ and monocyte cell populations did not quantitatively differ between the WT-Sham and TCRγδ^{-/-}-Sham animals (**Fig.**

2b-d), we agree with the reviewer that their function in a genetically modified $\gamma\delta$ T cell knockout model may indeed interfere with the observed outcomes. Daglas *et al.* (PMID 31665632) previously investigated the effects of CD4 and CD8+ T cells after TBI showing that the deficiency or pharmacological depletion of CD8+ T cells rather than CD4+ T cells improves neurological outcomes. Makinde *et al.* showed that monocyte depletion at the time of injury preserves white matter integrity (PMID 30383765) and improves functional outcomes (PMID 28993515). In our study, we demonstrated that CD4+, CD8+ and monocytes are decreased in TCR $\delta^{-/-}$ TBI (**Fig. 2b-d**). Ablation of V γ 4 cells using anti-V γ 4 and to some extent total $\gamma\delta$ T cells using anti-TCR $\gamma\delta$ resulted in reduction of CD4+, CD8+ and monocytes (**Fig. 4b-d**). Likewise, adoptive transfer of V γ 4 $\gamma\delta$ T cells resulted in increase in CD4+ T cells and CCR2+ monocytes (**Fig. 5b, e**). Due to the variety of subsets among CD4+, CD8+ and monocytes, investigating these cells in TCR $\delta^{-/-}$ knockout model after TBI requires a separate set of comprehensive experiments to be able to confidently derive robust conclusions. As a matter of fact, we show that *Foxp3* was increased in TCR $\delta^{-/-}$ mice (**Extended Data Fig. 2a**) whereas ablation of total $\gamma\delta$ T cells using anti-TCR $\gamma\delta$ (**Extended Data Fig. 5b**) as well as adoptive transfer of any $\gamma\delta$ T cell subset (**Extended Data Fig. 6b**) did not affect *Foxp3* expression, suggesting an expansion of FoxP3-expressing CD4 Treg cells in TCR $\delta^{-/-}$ TBI mice. For this reason and given the multitude of CD4+, CD8+ and monocyte subtypes, we believe that this is beyond the scope of our research study and further studies are needed to investigate the differentiation, maturation and function of each subset that might be impacted in the generation of the TCR $\delta^{-/-}$ model. The use of a genetically modified TCR $\delta^{-/-}$ model was added as a study limitation (please see elaboration under Reviewer 2, point #6).

Reviewer #2:

This experimental study by Hadi Abou-El-Hassan *et al.* is well designed and of high interest for the readership of Nature Communication. It addresses fundamental and so far unknown immune mechanisms in the case of traumatic brain injury (TBI) reflecting still a high scientific and clinical challenge. As already stated in the introduction part of the manuscript, there is almost no knowledge about the role of $\gamma\delta$ T cells in the context of traumatic TBI, resulting in a current lack of therapeutic strategies. A hypothesis was not defined. The aim of the study was to investigate the $\gamma\delta$ T cells and provide corresponding mechanisms involved in the acute (days) and chronic (year) immune response post TBI. Whether these findings lead to novel therapeutic approaches remains speculative.

Summary of the key results:

The study demonstrates that $\gamma\delta$ T cells drive and modulate the pathophysiology of TBI and the neurological functions as early as one day and up to one year. The clinically relevant mouse model of a controlled cortical impact revealed in TCR $\delta^{-/-}$ mice signs of reduced tissue damage and reduced inflammatory response accompanied by an improvement of a variety of neurocognitive functions. Furthermore, a balanced opposing response is suggested for V γ 1 and V γ 4 $\gamma\delta$ T cell subsets with their corresponding mediator pattern during TBI progression. An adoptive transfer of V γ 1 and V γ 4 $\gamma\delta$ T cell subsets from either WT mice or TCR $\delta^{-/-}$ mice at the time of injury revealed differential effects. Furthermore, the tight interaction of $\gamma\delta$ T cell subsets with the microglia is demonstrated on a functional and transcriptomic level, and suggest, that V γ 1 $\gamma\delta$ T cells reduce microglia activation and TBI-associated neuroinflammation.

Materials and Methods and data presentation: An established TBI mouse model was used and includes sufficient references for the chosen parameters. It contains all essential control animals and appropriate biological and technical replicates resulting in reproducible and reliable data. The methods used in the study are accurate, diverse and well- described in the method

section. Appropriate statistical methods were used to evaluate the data. Data-presenting graphs are well-designed and captured, figure legends include further information for clear understanding. All in all, the manuscript presents a well-structured study of high scientific value.

There are only some minor concerns:

1. The authors state in the abstract and introduction part that the role of $\gamma\delta$ T cells is “unknown” or “has not been explored” in the context of TBI. However, although little is known there is for example one study also on a murine TBI model, which addressed gd T cells at least in the spleen after TBI (e.g. Richard P Tobin et al. : Traumatic brain injury causes selective, CD74-dependent peripheral lymphocyte activation that exacerbates neurodegeneration. *Acta Neuropathologica Communications* 2014, 2:143). The authors should reword their statements a little more careful.

Response: We thank the reviewer for the observation. In their paper, Tobin *et al.* (PMID 25329434) showed a significant increase in splenic total $\gamma\delta$ T cells at 24 hours after injury. In our study, we have investigated $\gamma\delta$ T cells in acute as well as chronic TBI. We have edited the wording more carefully to add the word “largely” on line 20 and change the statement on lines 83-84 to “the contributions of $\gamma\delta$ T cells to the pathogenesis of TBI have not been explored”.

2. The authors also consider the $\gamma\delta$ Tcells the “first line of defense” (page 4, line 72). Commonly, neutrophils and monocytes and the complement system are defined the “first line of defense” so the authors should rather introduce $\gamma\delta$ T cells as contributing “part of the first line of defense”.

Response: We thank the reviewer for the note. We have reworded the statement on line 72 to clarify that $\gamma\delta$ T cells are “part of the first line of defense”.

3. Have the authors defined histological changes beyond only determination of the “lesion volume” (which was significantly reduced in the TCR $\delta^{-/-}$ mice)? It would be helpful for clinical translation to patho-morphologically assess at an early time point e.g. the degree of hematoma, neutrophil influx, fibrin deposition, monocyte/macrophages infiltration, thrombi formation, and especially signs of edema? In principal, these morphological findings could also be different in the TCR $\delta^{-/-}$ mice early after TBI?

Response: We thank the reviewer for the suggestion. On histology, we measured lesion volume (**Fig. 1a**) as well as microglial iNOS, microglial morphology, APP, p-Tau (**Fig. 2g-i**). Although hematoma size and fibrin deposition are less commonly evaluated in TBI, we agree with the Reviewer on the importance of evaluating the pathological findings in further details. Regarding monocytes, this has been assessed by flow cytometry on pericontusional brain tissue (**Fig. 2d-e**). Regarding neutrophils, we have conducted an experiment to evaluate neutrophilic infiltration of the CNS at day 2 after injury using the below experimental groups:

- WT-TBI ($n = 4$)
- TCR $\delta^{-/-}$ -TBI ($n = 4$)

We found that the infiltration of Ly6G+ neutrophils is also reduced in the ipsilateral injured cortex of TCR $\delta^{-/-}$ -TBI mice compared to WT-TBI. The following was added to the results section line 120:

“CD11b+Ly6G+ neutrophils (Extended Data Fig. 1b)”. We agree with the reviewer on the importance of evaluating brain edema which is indeed commonly studied in TBI. We conducted an experiment to measure water content using the below experimental groups:

- WT-TBI ($n = 6$)
- TCR $\delta^{-/-}$ -TBI ($n = 6$)

We measured brain edema at 3 days after injury and found significant reduction in brain edema of the ipsilateral hemisphere of TCR $\delta^{-/-}$ compared to WT. The result was added to **Fig 1**. The following was added to the Methods section on line 571: “**Brain edema:** Brains were removed at 72 hours after CCI and the ipsilateral hemisphere was weighed (wet weight). The hemisphere was then dried at 60 °C for 48 hours, and dry weights were obtained. The percentage of brain water content was expressed as (wet-dry weight)/wet weight \times 100 as previously described¹⁰⁷”. The following was added to the legend of **Fig. 1**: “and brain edema (right) at 3 days after TBI (WT-TBI $n = 6$, TCR $\delta^{-/-}$ TBI $n = 6$)”. The following was added to the Results section line 101: “We also assessed the percentage of brain edema and found significant reduction in brain edema of the ipsilateral hemisphere in the TCR $\delta^{-/-}$ group at 3 days post-CCI (**Fig. 1b**)”.

4. A graphical abstract -if agreed upon by the editor- could be very helpful for the reader to get these comprehensive data with many transcriptomic and adoptive transfer data condensed to the most important insights.

Response: We thank the reviewer for the suggestion. We have created a graphical abstract and is now included in **Extended Data Fig. 10**. The following was added to the Legends sections: “**Extended Data Figure 10. Graphical abstract summarizing the study findings. (a)** V γ 1 $\gamma\delta$ T cells secrete TGF- β promoting a homeostatic microglial function whereas V γ 4 $\gamma\delta$ T cells secrete IFN- γ and IL-17 resulting in neuroinflammation after TBI”.

Editorial Note: Figure created with BioRender.com.

5. Mislabeling: page 12 line 264: Fig. 6 i-n should read Fig. 5 i-n.

Response: We thank the reviewer for noting the mistake and it has been corrected in the revised manuscript.

6. Could the authors please provide in the discussion part some information on the limitations of the study or are there none?

Response: We thank the reviewer for the question. Our study has some limitations, and the following was added towards the end of the discussion section, line 428: "Our study reveals previously unknown roles of V γ 1 and V γ 4 $\gamma\delta$ T cell subsets in experimental TBI (**Extended Data Fig. 10a, b**). However, it is important to mention some few limitations. First, the use of the genetically modified TCR $\delta^{-/-}$ model may interfere with the maturation of other lymphoid and myeloid cells and contribute to the observed effects seen in **Figures 1-3**. Moreover, it may be argued that the lack of $\gamma\delta$ T cells in all tissues from birth as well as the baseline behavioral differences complicates our interpretation of the behavioral phenotype after TBI. Second, we found some differences in behavioral and immunologic effects between TCR $\delta^{-/-}$ mice and mice treated with anti-TCR $\gamma\delta$ depleting mAb. This could be related to the fact that the dose of the depleting antibodies we used was based on what has been reported in literature rather than an in-house gradient-guided optimal concentration selection. Moreover, it has been shown that the anti-TCR $\gamma\delta$ antibody clone UC7-13D5 may cause TCR internalization⁹³. Thus, it is possible that although the frequency of $\gamma\delta$ T cells is reduced in mice treated with anti-TCR $\gamma\delta$ mAb, these cells may still be functional at some extent. Third, while V γ 4 recipients demonstrated a worsened neuroinflammatory milieu, the lack of any phenotypic effect may be attributed to the chosen time point or the number of adoptively transferred cells. Lastly, the use of the designed lentivirus may inhibit the expression of IFN- γ R1 on all brain CD11b-expressing cells and not only microglia."

Reviewer #3:

In this manuscript, Habou-El-Hassan et al report an opposite role for V γ 1 and V γ 4 gd T cells in a model of Traumatic Brain Injury (TBI). Using neutralizing monoclonal antibodies and adoptive cell transfer, the authors demonstrate that the V γ 4 subset promotes TBI pathology through the production of IL-17 and IFN-g, while the V γ 1 subset appears protective, maintaining the microglia homeostasis through the production of TGF-b. Although mostly expected given the emerging roles of gd T cells in neuroinflammation, these findings are important and point at potential target for the development of immunotherapeutic strategies for the treatment of TBI and related pathologies. There are however a number of major shortcomings regarding the strength of some data and associated mechanisms, as well as the organization of the manuscript in general. The following issues should be addressed in order to ensure that the study is suitable for publication.

1- One of the main caveat of the study is the fact that the authors did not use littermates controls, especially for behavioural analysis. I could not find mention of littermates or at least, co-housing equilibration (>4 weeks) for the KO animals. When possible, mice should be derived from a Het X Het cross and WT littermates should be the appropriate controls. The importance of using WT littermates with as controls is crucial because there could be subtle differences in mouse housing conditions, handling and background genetics that may impact on overall immune profile at steady state and upon inflammation.

Response: We agree with the reviewer regarding the importance of using littermate controls. Other studies such as Wu *et al.* (PMID 34753914) used littermate controls for a subset of animals

with a C57BL/6NJ background (though not for those with a C57BL/6J background). While we did not use littermate controls since the TCR $\delta^{-/-}$ animals are derived from the same genetic background as the WT group, we did allow the animals to acclimate for 3-4 weeks prior to any treatment or behavioral testing. The following was added to the Methods section line 719: "Mice were allowed to acclimate for 3-4 weeks prior to testing".

2- The mice used are all males. Would the authors expect to reproduce their findings with female mice? Again, there may be subtle sex biased differences in animal behavior that should be at least discussed. Also, how were the group set? Are all the mice going through all the behavior assays? If so, in which order? This should be specified as this could also induce confounding variables, the mice potentially becoming less anxious along the sequence of the tasks. While some tests completely make sense in the context of TBI and current literature, the rationale for using some others and their resulting data are obscure to me. For example, why running the marble burying test? Can the authors comment on the higher compulsive behavior observed in TCRdKO one year post TBI? Some differences can be observed between Sham WT and TCRdKO, consistently with the previously described pro-anxiety role of gd T cells (de Lima et al, 2020). The authors should acknowledge these basal differences and discuss how they would (or not) be conserved upon TBI.

Response: We thank the reviewer for the detailed questions. Animals were brought into the behavioral rooms at least 30 minutes prior to testing and animals were tested consecutively, one from each group, rather than one group at a time. Behavioral apparatuses were cleaned between each animal. We performed behavioral testing from the least stressful to the most stressful as previously described (PMID 35002648). The following was added to the Methods section line 719: "At the day of testing, animals were acclimatized to the behavioral rooms for at least 30 minutes. Animals from different groups were tested consecutively. For the depletion and adoptive transfer experiments, mice underwent all behavioral assays beginning with the rotarod test, followed by the open field test and then the Morris water maze test".

We agree with the reviewer that the marble burying test is less commonly performed in experimental TBI. However, the marble burying test has been gaining more attention among the TBI community in more recent studies (PMIDs: 33461596, 35036685, 35002648, 27149139) as well as in ischemic brain injury (PMIDs: 30500429, 34139173) for the evaluation of compulsive behavior. The following was removed from line 111: "a standard test for" and was replaced with "used to evaluate".

As the reviewer mentions, it is indeed of utmost importance to understand and acknowledge the baseline differences between WT-Sham and TCR $\delta^{-/-}$ -Sham mice. We agree that, unfortunately, these baseline differences may confound our derived conclusions for which we later decided to employ two useful experimental paradigms: depletion and adoptive transfer. The following was added to the study limitations paragraph (reviewer 2, point #6) on line 428: "It is worth mentioning that an increased compulsive behavior and a tendency towards a decreased anxiety-like behavior was noted in TCR $\delta^{-/-}$ -Sham compared to WT-Sham mice, consistent with recent findings³⁸. It may be argued that the lack of $\gamma\delta$ T cells in all tissues from birth as well as the baseline behavioral differences complicates our interpretation of the behavioral phenotype after TBI".

We agree with the reviewer on the importance of investigating the impact of sex on the favorable outcomes observed in male TCR $\delta^{-/-}$ animals. To address a potential sex effect on the observed phenotype, we conducted behavioral testing (rotarod, open field, MWM) in female mice at 1 month after TBI using the below groups:

- Female WT-Sham ($n = 8$)
- Female WT-TBI ($n = 12$)
- Female TCR $\delta^{-/-}$ -Sham ($n = 8$)

- Female TCR $\delta^{-/-}$ -TBI ($n = 10$)

The following was added to the results section, line 114: “Given the recent evidence on divergent sex responses at the molecular⁵¹ as well as behavioral⁵² levels after TBI⁵³, we investigated whether female TCR $\delta^{-/-}$ animals confer comparable behavioral neuroprotection to their male counterparts. We found no significant differences at the behavioral level between WT-TBI and TCR $\delta^{-/-}$ -TBI female mice at 1 month after injury (**Extended Data Fig. 1a**). We also found that, except for anxiety-like behavior, TBI itself did not significantly affect cerebellar function or memory between WT-Sham and WT-TBI female mice. This may be explained by the dampened acute neuroinflammation in females that has been proposed to be driven by the neuroprotective effects of estrogen and progesterone⁵⁴”. The following was added to line 436: “and female”. The following was added to line 436: “In each figure, male mice were used unless otherwise specified in the legend”.

3- The authors observe a sizeable population of Vg1-Vg4- T cells, without commenting on it. Do the frequency of this subset vary upon TBI? Are these cells Vg6+? Or even Vg7+, referring to Benakis et al, 2016? Then, would they be influenced by the microbiota? In this same line, what is the population of CD11b+Ly6Cint cells that clearly appears in every dot plots, from Figure 2 to Figure 5, and seems to vary according to the conditions and models used? Can they authors quantify and comment on these? More generally, the cell frequencies obtained by FACS analysis would be greatly strengthened by displaying associated absolute cell counts.

Response: We thank the reviewer for the comments. The absolute cell counts generally correlated with the shown percentages in our experiments. While the role of the microbiome is out of the scope of this study and although we did not investigate whether the Vg1-Vg4-population is Vg6 or Vg7 or any other $\gamma\delta$ T cell subset, we were able to temporally quantify the Vg1-Vg4-population. We found a significant decrease in the Vg1-Vg4- cell population in the brain at 3 days after TBI.

The following was added to the Results section line 213: “as well as a to-be-characterized Vg1-Vg4- cell population”. The following was added to the Discussion section line

399: “We found significant decrease in a yet undefined double-negative V γ 1-V γ 4- cell population in the brain at 3 days after TBI which requires further characterization”. The legend of **Extended Data. Fig. 4** was edited accordingly.

The functional distinction between Ly6C^{hi} and Ly6C^{int} (interchangeably named Ly6C^{lo}) monocytes especially in the brain remains poorly understood. We plotted Ly6C^{int} monocytes that generally correlated with Ly6C^{hi} monocytes.

The following was added to the Results section lines 249 and 296: “and Ly6C^{int}”. The legends of **Figures 2 and 4-7** were edited accordingly.

4- Previous studies report a dual role for IL-17 in a model of stroke, triggering the onset of neuroinflammation (Benakis et al, 2016) while promoting neural recovery after 14 days (Lin et al, 2016). In such context, it would be interesting to discuss potential windows on which these different gd T cell subsets could intervene and regulate TBI time-course and severity. As proof of concept, different timepoint of in vivo cell transfers or manipulation of the microglia cultures could have been tested.

Response: We thank the reviewer for the insightful comment. Investigating neurogenesis in TBI is a specialized field and requires a separate set of *in vivo* and *in vitro* experiments to confidently determine the specific $\gamma\delta$ T cell subset that produces a specific to-be-determined pro-neurogenesis cytokine. Performing *in vivo* cell transfers or manipulation of microglia cultures may not be sufficient to derive robust conclusions without further comprehensive experiments using various knockout models of both cytokines of interest as well as that of the neuronal stem cells. Interestingly, Lin *et al.* showed that astrocytes are the major source of IL-17A in the delayed phase of stroke rather than $\gamma\delta$ T cells. We agree with the Reviewer that determining the optimal windows of intervention in the acute vs. subacute vs. chronic phases is important and is currently a future goal of our laboratory. The following was added to the Discussion section line 424: “It was shown that accumulation of meningeal IL-17+ $\gamma\delta$ T cells was associated with increased size of the acute infarct⁹¹ whereas increased levels of IL-17A at 14 days after stroke promoted neural recovery⁹² suggesting paradoxical and time-sensitive effects of IL-17 in brain injuries”.

5- Regarding the in vivo cell transfer experiments, it would be interesting to isolate the V γ 1 and V γ 4 subsets from TBI mice, besides naïve animals. Are they expected to display a different outcome? In other words, how would their functional properties being shaped by the TBI environment? Can the authors speculate on where and how would they be primed? In this line, don't V γ 1 also produce IL-17 and IFN-g upon TBI? Why not? What could make switch their typical type 1 effector program towards the secretion of TGF-b?

Response: We thank the reviewer for the comment. We believe that V γ 1 do not necessarily switch into secreting TGF- β as they already express more LAP compared to V γ 4 (**Extended Data Fig. 6c**) as well as TGF- β (**Extended Data Fig. 7a**) at baseline. We then became interested in utilizing this finding as a potential treatment for the post-traumatic neurocognitive deficits (**Fig. 5i-n**). Accordingly, we discussed the homeostatic role of V γ 1 cells in the Discussion lines 403-406: “Although V γ 1 $\gamma\delta$ T cells can produce pro-inflammatory cytokines including IFN- γ and IL-17, which we found to be involved in V γ 4 cell-mediated neuroinflammation post-TBI, it was possible that a subset of regulatory V γ 1 cells also infiltrated the brain and regulated inflammation” which was further supported by lines 409-412: “Consistent with this, we found that neuroprotection conferred by V γ 1 $\gamma\delta$ T cells following TBI was abrogated in the absence of V γ 1+LAP^{hi} cells *in vivo* and after TGF- β neutralization *in vitro*, indicating that LAP-expressing V γ 1 $\gamma\delta$ T cells were responsible for the beneficial effects of V γ 1 $\gamma\delta$ T cells in TBI”. We agree with the reviewer on the importance of isolating V γ 1 $\gamma\delta$ T cells from TBI besides naïve mice. For this, we conducted an experiment in which we quantified IFN- γ and IL-17 secretion by V γ 1 cells isolated from the deep cervical lymph nodes and spleen of Sham and TBI mice 3 days after injury using the below two groups:

- WT-Sham ($n = 3$)
- WT-TBI ($n = 3$)

We found that V γ 1 $\gamma\delta$ T cells secrete as much IFN- γ as V γ 4 cells. While V γ 1 cells reactively produce more IFN- γ after TBI in both the deep cervical lymph nodes and spleen, the secretion of IL-17 was not significantly different between sham and TBI. Interestingly, we discovered that V γ 1 cells secrete little to no IL-17. For instance, lymph node V γ 1+ cells stained for 0.00%, 0.06% and 0.05% IFN γ -IL17+ cells for the three Sham samples.

The following was added to the Results section line 292: “Of note, we found that V γ 1 $\gamma\delta$ T cells isolated from the deep cervical lymph nodes and spleen reactively produced more IFN- γ at 3 days after TBI, however, V γ 1 cells were found to secrete little to no IL-17 (**Extended Data Fig. 6d, e**)”. The legend of **Extended Data. Fig. 6** was edited accordingly.

6- Can the authors comment on how would V γ 1 or V γ 4 influence BBB permeability? Could they use an alternative and functional way to characterize it? How V γ 1 infiltrate the brain if the BBB is intact?

Response: We thank the reviewer for the comment. We employed the controlled cortical impact (CCI) model of TBI as previously described (PMID 23329160) which is highly reproducible as it allows to specify the injury diameter, depth as well as dwell time and impact velocity. The CCI model results in a contusion on the brain as shown in **Fig. 1a** which impairs the blood-brain barrier (BBB) causing edema and leukocyte infiltration (PMID 28910616). Therefore, in the CCI model, the blood-brain barrier is no longer intact, and we show that its healing is facilitated via V γ 1 $\gamma\delta$ T cells (**Fig. 5h**). In response to point #3 of Reviewer 2, we conducted an experiment to evaluate brain edema by weighing brain tissue before and after drying at 60 °C for 48 hours.

7- Figure 3 is really difficult to understand. It needs to be reorganized, as it is hard to follow the panel numbering. For a clarification purpose, some data could be transfer to supplementary information. In general, if Vg4 and Vg1 have opposing role, it is not very intuitive to analyze the microglia phenotype in the presence or absence of total gd T cells, using a full KO strategy. I understand the general unbiased screening approach but this should rather be presented in supplementary data, with some data being then exploited/validated in mice treated with anti-Vg4 and anti-Vg1, as done in Fig 4o. On the hand, Extended Data Fig. 3g and h would be important to appear in the main figure.

Response: We thank the reviewer for the comment. Organizing Figure 3 was challenging to us and we have tried our best to organize the panels prior to the initial submission to best track the panel numbering with a predominantly vertical fashion. Since the unbiased screening approach was done on different genetic backgrounds, the resulted differentially expressed genes were different among the experimental approaches which forced us to display each approach in separate figures. We agree with the reviewer to make the figure less crowded, and we have decided to remove Fig. 3n (the circular plot), keep Extended Data Fig. 3g and h due to space restrictions, while keeping Fig. 3j in the main figure representing the main genes.

8- Overall, it is difficult to follow the track of the genes that are highlighting from one figure to another, referring to the neuroinflammatory / homeostatic microglial status. Also, more explanations should be given to establish the link between this microglial status post-TBI and the other neurodegenerative conditions (line 165 to 172). I do not understand how to define “a chronic TBI-associated microglia signature” by linking the TBI signature to the one observed in other neurodegenerative diseases. Then, what about the association between the “acute TBI microglial transcriptome” and AD, PD, ALS (Fig 3n)?

Response: We thank the reviewer for the comment. Since epilepsy, aging, AD, PD and ALS are, by large, chronic long-term processes, we decided to define a TBI-associated microglia signature in the chronic phase only. The chronic TBI microglia transcriptomic signature (as listed in the original **Supplementary Table 1**) was derived by excluding the overlapping genes between TBI

Key differentially expressed microglia genes	
Homeostatic/ Regenerative	Inflammatory/ Neurodegenerative
Tmem119	Ccl2
Fkbp5	Casp1
Lrrc3	Tlr4
Siglech	Cd86
Tgfb1	Hmgb1
Fcrls	Stip1
Atf3	Il1b
Nfkbia	CD74
Dusp1	Ccr2
Egr1	Ccl5
Fam49b	Sod1
Socs4	Spp1
Irf2bpl	Ccl12

and the other neurological diseases. Nevertheless, we have decided to remove these comparisons from the manuscript and focus on $\gamma\delta$ T cells in TBI. Lines 165-172 were deleted and the legends of **Fig. 3** and **Extended Data Fig. 3** were edited accordingly. The “chronic TBI signature” column was removed from **Supplementary Table 1**. To make it easy on the reader to discriminate between homeostatic/regenerative and inflammatory/neurodegenerative microglia genes, a table of key differentially expressed microglia genes that we found in our study was added to panel b of a new **Extended Data Fig. 10** in addition to a graphical abstract suggested by point #4 of Reviewer 2.

Minor comments and proofreading:

- **Line 158: “Acute microglia modulation was predicted to be driven by both IFN- γ (z = -1.765) and IL-17A (z = -1.965) as upstream regulators (Fig. 3j)”**. This does not appear clearly. Response: We thank the reviewer for the comment. We have increased the size of the p value and z score and we will improve the resolution of the published figures.

- **Line 188: “We found that microglia from aged TCR $\delta^{-/-}$ mice had only 84 DEGs vs.1,879 DEGs in microglia from aged WT-TBI mice (Extended Data Fig. 3b-e)”** Unclear sentence that would benefit some rephrasing. Can they authors develop on the potential implications? Response: We thank the reviewer for the comment. The following was added to the Results section line 190: “suggesting the involvement of $\gamma\delta$ T cells in aging. For instance, aging-associated genes such as Clec7a^{58, 68} and Ccl2⁶⁹ found to be upregulated in aged WT-TBI mice were not differentially expressed in aged TCR $\delta^{-/-}$ TBI mice”.

- **Line 200: “To investigate whether V γ 1 and V γ 4 $\gamma\delta$ T cell subsets play opposing roles in acute and chronic TBI, we first performed a kinetic study”**. This would not define their roles. Please rephrase. Response: We thank the reviewer for the comment. The statement was deleted.

- **Line 394: “gd T cells expand upon TBI in lymphatics”**: Ref 86 does not match this statement, please cite the original paper. Response: We thank the reviewer for the comment. Reference 86 was removed and original articles (PMIDs: 26030524, 30224810, 32929273) were cited.

- **Lines 436 and 446: please refer to C57BL/6j not just C57BL/6 as it is the case in the rest of the material and method section.** Response: We thank the reviewer for the comment. C57BL/6 was corrected to C57BL/6J.

- **Line 647: typo “three 3” erase one of them.** Response: We thank the reviewer for the comment. “three” was deleted.

- **Line 672: “extended data fig4b” does not show the gating strategy for the Vy subset sorts.** Response: We thank the reviewer for the comment. The gating strategy has been added to **Extended Data Fig. 9**.

- **Line 852: the authors mention “BDP, breakdown products” that do not appear on the figure itself.** Response: We thank the reviewer for the comment. We apologize for the mistake and it has been deleted.

- **Extended Data Fig1:**

- **Typo TCT-/- that needs to be replaced by TCR.** Response: We thank the reviewer for the comment. "TCT" was corrected to "TCR".

- **Scales on the microcopy picture are missing.** Response: We thank the reviewer for the comment. Scales have been added to the figure.

- **Lines 859-861: sham are not shown, remove them from the legend.** Response: We thank the reviewer for the comment. They have been deleted from the legend.

- **Lines 870-872: sham are not shown, remove them from the legend.** Response: We thank the reviewer for the comment. They have been deleted from the legend.

- **Line 1021: sham are not shown, remove them from the legend.** Response: We thank the reviewer for the comment. They have been deleted from the legend.

- **Fig4g and Fig5g: scales are missing.** Response: We thank the reviewer for the comment. Scales have been added to the mentioned figures as well as **Fig. 2j**, **Fig. 6g** and **Fig. 7g**.

REVIEWER COMMENTS

Reviewer #1 (Remarks to the Author):

This revised manuscript has largely addressed the reviewer's questions and concerns.

Reviewer #2 (Remarks to the Author):

The authors have addressed all the reviewers' concerns and thereby could significantly improve the manuscript for the readership of Nature Communications. The additional experiments with the extended data pool confirm the previous conclusions drawn by the authors and overall strengthen the manuscript. The graphical abstract (fig. 10) transfer the novel message in a straightforward and reasonable manner.

Reviewer #3 (Remarks to the Author):

The authors did a very good job in answering most part of the comments that were raised by the reviewers, which greatly improved the quality of their manuscript. I appreciate their new extended discussion, namely the paragraph regarding the limitations of their study.

The authors also added important pieces of experimental data, addressing their conclusions in female mice, as well as documenting on the neutrophilic infiltration and absolute cell count.

Finally, they made a great effort in the reorganisation of their bioinformatics data, which will undoubtedly help the reader.

Notwithstanding, my main concern remains regarding the fact that the authors did not use (co-housed) littermates controls (even if the WT controls used as from the same genetic background). This point is absolutely critical, especially for behavioural analysis, and especially in the context of TBI.

TBI is well known to impact the microbiome composition, and the respective correlate may be foreseen. Thus, as TCRd^{-/-} and the commercial WT group (even sharing the same genetic background) are expected to have a different microbiota, the results observed could be partially attributed to particular gut commensals, and could be related or not to the presence of total gd T cells.

In my opinion, this is a major caveat of the study, which needs to be addressed.

Vy1 and Vy4 $\gamma\delta$ T cell subsets play opposing roles in traumatic brain injury

Reviewer 3. The authors did a very good job in answering most part of the comments that were raised by the reviewers, which greatly improved the quality of their manuscript. I appreciate their new extended discussion, namely the paragraph regarding the limitations of their study. The authors also added important pieces of experimental data, addressing their conclusions in female mice, as well as documenting on the neutrophilic infiltration and absolute cell count. Finally, they made a great effort in the reorganization of their bioinformatics data, which will undoubtedly help the reader.

Notwithstanding, my main concern remains regarding the fact that the authors did not use (co-housed) littermate controls (even if the WT controls used as from the same genetic background). This point is absolutely critical, especially for behavioral analysis, and especially in the context of TBI. TBI is well known to impact the microbiome composition, and the respective correlate may be foreseen. Thus, as TCRd^{-/-} and the commercial WT group (even sharing the same genetic background) are expected to have a different microbiota, the results observed could be partially attributed to particular gut commensals and could be related or not to the presence of total gd T cells. In my opinion, this is a major caveat of the study, which needs to be addressed.

Response: We agree with the reviewer regarding the importance of using littermate controls in behavioral testing after brain injury, but we strongly disagree that it is crucial for the validity of our study and precludes publication of our revised manuscript in the present form. Please note that the point regarding littermate controls was not raised by either Reviewer #1 or Reviewer #2. More importantly, other publications on TBI did not use littermate controls including the following: Schiweck, et al Nature Communications (2021), Wu, et al Cell Death and Disease (2021) and Rubenstein, et al Acta Neuropath Comm (2017).

If we were to respond to Reviewer #3's request regarding littermate controls, it would require 6 months of breeding and another 6 months to repeat all the experiments

to test both acute and chronic timepoints in TBI, which we do not feel is scientifically justified. We asked senior faculty in our center to give an opinion regarding the need for a year's worth of experiments for this manuscript. All felt that the scientific findings were valid and could be published without further experimentation. Nevertheless, we would like to provide information on some behavioral experiments in which we used TCRd^{-/-} ($\gamma\delta^{-/-}$) from homozygous breeders and TCRd^{-/-} from heterozygous breeders and the littermate controls. In the marble burying test shown here, TCRd^{-/-} mice from homozygous or heterozygous breeders behaved similarly to their counterpart controls (left panel: naïve WT mice were purchased from the Jax labs; right panel: littermate WT ($\gamma\delta^{+/+}$) and littermate heterozygous ($\gamma\delta^{+/-}$) were bred in-house, which means that offspring used came from the same cage (co-housed)). Although these experiments do not rule out the fact that the behavioral observations shown in our TBI

studies would be similarly between TCRd-/- and littermate controls, it provides strong evidence that this would likely be the case. Please note that these experiments are part of another project and will be soon send for publication.

Thus, we are resubmitting the manuscript with the current data and a statement in the discussion that acknowledges that a potential limitation of our study is that littermate controls were not used (statement provided in red).

Thank you for your careful consideration and support of our work. We hope you will find our revised manuscript acceptable for publication.

REVIEWERS' COMMENTS

Reviewer #3 (Remarks to the Author):

This revised manuscript has addressed most the reviewer's questions and concerns. I namely appreciate the new paragraph discussing the weight of the microbiome in this context.

Vy1 and Vy4 $\gamma\delta$ T cell subsets play opposing roles in traumatic brain injury

Reviewer #3

This revised manuscript has addressed most the reviewer's questions and concerns. I namely appreciate the new paragraph discussing the weight of the microbiome in this context.

Response: We thank the reviewers and the editorial team for reviewing our re-submitted work. All the reviewers' concerns have been addressed.